



# Estimating irrigation water use over the contiguous United States by combining satellite and reanalysis soil moisture data

Felix Zaussinger[1], Wouter Dorigo[1], Alexander Gruber[1,2], Angelica Tarpanelli[3], Paolo Filippucci[3], and Luca Brocca[3]

[1]CLIMERS - Research Group Climate and Environmental Remote Sensing, Department of Geodesy and Geoinformation, TU Wien, Vienna, Austria
[2]Department of Earth and Environmental Sciences, KU Leuven, Heverlee, Belgium
[3]Research Institute for Geo-Hydrological Protection, National Research Council, Perugia, Italy

**Correspondence:** Felix Zaussinger (felix.zaussinger@geo.tuwien.ac.at) and Wouter Dorigo (wouter.dorigo@geo.tuwien.ac.at)

**Abstract.** Effective agricultural water management requires accurate and timely information on the availability and use of irrigation water. However, most existing information on irrigation water use (IWU) lacks the objectivity and spatio-temporal representativeness needed for operational water management and meaningful characterisation of land-climate interactions. Although optical remote sensing has been used to map the area affected by irrigation, it does not physically allow for the

5 estimation of the actual amount of irrigation water applied. On the other hand, microwave observations of the moisture content in the top soil layer are directly influenced by agricultural irrigation practices, and thus potentially allow for the quantitative estimation of IWU. In this study, we combine surface soil moisture retrievals from the spaceborne SMAP, AMSR2, and AS-CAT microwave sensors with modelled soil moisture from MERRA-2 reanalysis to derive monthly IWU dynamics over the contiguous United States (CONUS) for the period 2013-2016. The methodology is driven by the assumption that the hydrol-

10 ogy formulation of the MERRA-2 model does not account for irrigation, while the remotely sensed soil moisture retrievals do contain an irrigation signal. For many CONUS irrigation hot spots, the estimated spatial irrigation patterns show good agreement with a reference data set on irrigated areas. Moreover, in intensively irrigated areas, the temporal dynamics of observed IWU is meaningful with respect to ancillary data on local irrigation practices. State-aggregated mean IWU volumes derived from the combination of SMAP and MERRA-2 soil moisture show a good correlation with statistically reported state-level

15 irrigation water withdrawals but systematically underestimate them. We argue that this discrepancy can be mainly attributed to the coarse spatial resolution of the employed satellite soil moisture retrievals, which fails to resolve local irrigation practices. Consequently, higher resolution soil moisture data are needed to further enhance the accuracy of IWU mapping.



## 1 Introduction

The agricultural sector uses over $70\%$ of global freshwater withdrawals for irrigation (Shiklomanov, 2000; Foley et al., 2011). As a result of world population increase and rising living standards water will be a major constraint for agriculture in the coming decades. In addition, climate change will likely have a profound impact on irrigation demand throughout the world.

The projected increase in global mean temperature and changing precipitation patterns are expected to decrease natural water availability in already water-scarce regions of the world (Vörösmarty et al., 2000; Rockström et al., 2012; Kummu et al., 2016). For instance, Döll (2002) showed that around 2/3rd of the areas that were irrigated in 1995 will require more irrigation water by 2070. Moreover, predictions show that the hydrological cycle will intensify. Hence, drought and flood events are expected to occur both more frequently and severely, which further impairs water availability for agriculture (Allan and Soden, 2008).

On the other hand, irrigation itself is an important anthropogenic climate forcing (Sacks et al., 2009). It influences the surface water and energy balance through directly increasing soil moisture, which in turn modulates the partitioning of energy between sensible and latent heat (Seneviratne et al., 2010). Subsequently irrigation cools the land surface on local to regional scales through increasing evapotranspiration (ET), whereas the increased availability of atmospheric water vapor can enhance cloud

cover and precipitation (Boucher et al., 2004; Lobell et al., 2006; Sacks et al., 2009). Researchers agree that irrigation may have masked the full warming signal caused by greenhouse gas emissions (Bonfils and Lobell, 2007; Kueppers et al., 2007). As a past expansion of irrigated area and an overall increase in irrigation intensity may have significantly affected surface temperature observations, it is crucial to include irrigation impacts both in understanding historical climate and modeling future climate trends (Lobell et al., 2006). Assuming a similar expansion of irrigation as in recent decades, some regions may actually

benefit from this irrigation cooling effect. As outlined in Ozdogan et al. (2006), ET and in turn irrigation water requirements can decrease within agricultural micro-climates. However, non-linear repercussions on temperature extremes can be expected when the required water supply cannot be met (Thiery et al., 2017) and (semi-)arid regions are generally expected to be adversely affected by water scarcity (Kueppers et al., 2007). As a consequence, especially in water-scarce regions, government agencies and water managers are challenged to increase water use efficiency, optimise the distribution of water among farms,

and detect illegal groundwater pumping activities (Siebert et al., 2010; Taylor et al., 2012). For example, as a consequence of prolonged winter precipitation deficits and positive temperature anomalies from 2012-2017 a record breaking drought peaking in 2015 affected the California Central Valley. While farmers tried to compensate the 2015 surface water shortage by pumping more groundwater, a net water shortage of over $3\,\mathrm{km}^3$ resulted in the fallowing of approximately $230\,000\,\mathrm{ha}$ of land (Howitt, 2015).

To date, irrigation practices are typically not explicitly included in land surface, climate, or weather models. On the other hand, irrigation directly impacts land surface temperature, humidity, and soil moisture observations, and through them indirectly impact model simulations when they are being assimilated (Tuinenburg and Vries, 2017). In recent years, a range of climate modelling studies employed irrigation modules, which were mainly based on a combination of static) spatial maps of



irrigated area and soil moisture and/or vegetation data to approximate seasonal IWU (Lobell et al., 2006; Bonfils and Lobell, 2007; Kueppers et al., 2007). However, the simulated impact of irrigation on both global and regional climate showed considerable variation across studies. Sacks et al. (2009) argued that this discrepancy is primarily explained by systematic differences in the control of irrigation water application within the respective modules, e.g. by climate, food demand, and economical

conditions. Also, fields may be either over- or under-irrigated with respect to the physically "ideal"amount. Hence, current irrigation modules are unable to consistently reflect real-world conditions and thus introduce uncertainties in the model simulations. Consequently, information on the spatio-temporal distribution and development of actual IWU is needed to improve the representation of land-atmosphere feedbacks in model simulations (Ozdogan et al., 2010a).

## 1.1 Statistics on irrigated areas and water withdrawals

Available information on irrigated areas and particularly irrigation water use lacks objectivity, spatial consistency and temporal resolution needed for large-scale hydrological assessments and modeling (Deines et al., 2017). On local to regional scales, some irrigation districts conduct regular surveys, but often the data are not publicly available, lack geo-referencing and are difficult to compare between regions due to different sampling techniques. The elementary source of large-scale irrigation data are national and sub-national statistical units, which in most countries routinely collect information on irrigated area and/or

irrigation water withdrawals. Data are usually represented as area equipped for irrigation (AEI), and in some cases also reflect the area actually irrigated (AAI) in the respective year of the census (Siebert et al., 2005)). The Global Map of Irrigation Areas (GMIA) was the first global-scale geospatial irrigated area data set (Döll, 2002; Döll and Siebert, 2002) based on such statistics. GMIA combines sub-national irrigation data from various sources (FAO, UN, World Bank, Agriculture Departments) and geospatial information on the location and extent of irrigation schemes (point-, polygon- and raster data, land cover maps,

and satellite imagery) to map AEI and AAI at $0.5°$ resolution around the year 2000. In subsequent versions the resolution was improved to $5'x5'$ (Siebert et al., 2005, 2007). However, the large variability in the quality of the underlying statistical inventory data is propagated into the uncertainty of the final spatial map (Siebert et al., 2005).





In summary, the main limitations of statistical inventories and derived products are:

1. The quality of the data varies significantly among countries (Siebert et al., 2010). While for instance the United States agricultural census is considered to have high quality, many developing countries lack the resources for comprehensive reporting.

2. National statistics are usually only valid for single years and depend on the individual compilation cycle of each country (e.g. every 5 years in case of the US).

3. Irrigated area estimates usually reflect areas equipped for irrigation, rather than areas actually irrigated. Depending on climatic and market conditions, farmers may decide to only cultivate and irrigate a portion of their fields.

4. Irrigation volume estimates reflect irrigation water withdrawals rather than actual irrigation water use (e.g. if rainfall is sufficient, already withdrawn spare water is stored in reservoirs instead of being irrigated)

5. Naturally, survey-based statistics are only based on a sample of farms, which may not be representative.

6. Conventional methods are unable to reflect illegally withdrawn water used for irrigation (Roseta-Palma et al., 2014; Saffi and Cheddadi, 2010).

## 1.2 Remote sensing for irrigation mapping

Remote sensing offers the potential to overcome the limitations of statistical inventories by providing synoptic and timely information of biogeophysical variables that are either directly or indirectly related to irrigation.

### 1.2.1 Optical and thermal remote sensing

Data acquired by optical sensors (AVHRR, MODIS, Landsat) have been extensively used to identify irrigated areas on local-, regional- and global scales. Vegetation indices have been identified as effective proxies for irrigation practices, because irrigated and non-irrigated cropland show different spectral responses during the peak growing season (Ozdogan et al., 2010b). A wide range of studies used vegetation indices to map annual irrigated areas and their changes through time, sometimes in combination with statistical inventory data.

Only few global land use-land cover (LULC) maps based on optical remote sensing separate irrigated from rainfed croplands. For example, the USGS Global Land Cover Characteristics (GLCC) data set was derived from 1 km Advanced Very High Resolution Radiometer (AVHRR) sensor data and identified four types of irrigated croplands in the year 1992 (Loveland et al., 2000). However, the classification algorithms used were not tailored to irrigated area mapping, thus resulting in low classification accuracies. Large discrepancies were found between USGS GLCC and country level reports of irrigated area, originating from both the uncertainties of the inventory data and technical limitations of the remote sensing data sets (Vörösmarty et al., 2000). Through a combination of unsupervised clustering and expert knowledge, the European Space Agency



(ESA) Climate Change Initiative (CCI) has produced global land cover product at 300 m resolution using Medium Resolution Imaging Spectrometer (MERIS) data (Bontemps et al., 2013). It distinguishes irrigated and non-irrigated cropland for 2000, 2005 and 2010, but the irrigated class is likely to be considered unreliable, as for instance no irrigated areas were mapped within the contiguous United States (CONUS).

Other studies used approaches specifically tailored to irrigated area mapping. For instance, the global data set of monthly irrigated and rainfed crop areas around the year 2000 (MIRCA2000) provides irrigated and rainfed areas for 26 crop classes for each month of the year at 5′ resolution (Portmann et al., 2010). For this purpose, agricultural census statistics, national reports, databases, a map of crop specific annual harvested area, a cropland extent map, the GMIA, crop calendars and ancillary infor-
mation on climate and topography were combined. Using quantitative spectral matching techniques on NDVI time series from multiple sensors (AVHRR, SPOT-1, MODIS, Landsat 7 and JERS-1 SAR) in combination with climate (monthly precipitation and temperature data from CRU) and ancillary data (GTOPO30 1 km DEM, global tree cover), the International Water Management Institute (IMWI) produced a Global Irrigated Area Map (GIAM) at 1 km resolution around the year 2000 (Thenkabail et al., 2009). More recently, Salmon et al. (2015) created a global map of rain-fed, irrigated and paddy croplands (GRIPC)
around the year 2005 at 500 m spatial resolution using supervised classification of remote sensing, climate, and agricultural inventory data. However, there are large discrepancies between the different global data sets (Salmon et al., 2015). Moreover, three of the four existing global maps (GMIA, MIRCA2000 and GRIPC) rely on agricultural inventory data for the classification of irrigated areas, which is subject to major limitations concerning quality and accuracy. In addition, the maps are limited to single years (GMIA, GIAM, GRIPC), or single months within a single year (MIRCA2000), thus not being able to address the
high inter-annual variability of irrigated areas, which is mainly governed by climate and market conditions (Deines et al., 2017).

On a continental scale, the MODIS Irrigated Agriculture data set for the conterminous United States (MIrAD-US) was created by assimilating county level irrigation statistics with MODIS-derived seasonal peak Normalised Difference Vegetation Index (NDVI) to spatially identify irrigated and non-irrigated lands at 250 m resolution (Ozdogan and Gutman, 2008; Pervez
et al., 2008; Pervez and Brown, 2010). A significant drawback is that the map compilation is tied to the same 5-year cycle of the United States Department of Agriculture (USDA) Census of Agriculture. Ambika et al. (2016) mapped irrigated areas from 2000–2015 at 250 m resolution over India by using 250 m MODIS seasonal peak NDVI data and 56 m LULC data. Teluguntla et al. (2017) used spectral matching techniques and automated cropland classification algorithms to infer cropland extent, irrigated versus rainfed croplands, and cropping intensities over Australia. The latter two products allow to study inter-annual
variability of irrigated areas (Ambika et al., 2016).

On a regional scale, higher resolution Landsat imagery was adopted by a range of studies. Ozdogan et al. (2006) used 30 m Landsat imagery to map changes in annual irrigated area from 1993 to 2002 in southeastern Turkey based on NDVI thresholding approaches and compared them with estimates of irrigation water requirements inferred from potential evapotranspiration.
In a recent study, Deines et al. (2017) produced annual irrigation maps for 1999-2016 for a region in the High Plains Aquifer



(United States) at 30 m resolution. Pun et al. (2017) used a combination of surface energy balance partitioning and vegetation indices to classify irrigated and non-irrigated croplands at 30 m resolution in Nebraska.

Thermal remote sensing has been widely used to map irrigation water based on estimating potential evaporation from surface
energy heat fluxes and the application of specific crop factors (Rosas et al., 2017). A well-known technique is the Surface Energy Balance Algorithm for Land, which estimates variables of the hydrological cycle based on remotely sensed surface energy balance components (Bastiaanssen et al., 1998). However, these methods are only able to provide estimates on irrigation water requirements (i.e. what the amount of water a plant would ideally need), as opposed to actually irrigated water, as in practice fields are often over- or under-irrigated.

**1.2.2  Microwave remote sensing**

Microwave observations are widely used to estimate soil moisture (Entekhabi et al., 2010; Wagner et al., 2013; Dorigo et al., 2017). Major advantages of microwave observations are their all-weather capability and the intrinsic capacity to sense a geophysical variable which is directly and physically linked to irrigation.

The first study to investigate the utility of satellite soil moisture retrievals for irrigation mapping was carried out by Kumar et al. (2015). They used soil moisture retrievals from ASCAT, AMSR-E, SMOS and Windsat, and the ESA CCI multi-satellite surface soil moisture product in combination with soil moisture estimates from the Noah LSM to map irrigated areas in the CONUS. Their key assumption was that irrigation is not included in the formulation of LSM, whereas satellite-derived soil moisture is expected to reflect the changes in soil moisture induced by irrigation. Based on synthetic data, they were able to
detect differences between the probability density functions of satellite and modelled soil moisture. However, the satellite data showed only few systematic differences that could be reliably related to irrigation practices.

Qiu et al. (2016) compared trends from 1996–2010 in China of ESA CCI, ERA Interim/Land reanalysis, and in-situ soil moisture, as well as precipitation. They observed significant discrepancies between precipitation and satellite soil moisture trends over irrigated areas, which they ascribed to irrigation. Escorihuela and Quintana-Seguí (2016) compared three global
satellite soil moisture products (ASCAT, AMSR-2 and SMOS) with model soil moisture estimates from the ISBA scheme within SURFEX (forced with meteorological data) in the Mediterranean. Only a downscaled version of SMOS (SMOScat) showed significantly lower correlations over irrigated areas. The authors argued that primarily due to the coarse spatial resolution of the native soil moisture retrievals the other products were not able to resolve the irrigation signal from the soil moisture
signal from the surrounding dry-land area. Very recently Lawston et al. (2017) investigated the potential of the new SMAP enhanced 9 km SM product to identify irrigation signals in three semi-arid regions in the western United States. Results showed that SMAP soil moisture carries a clear irrigation signal from rice irrigation in the Sacramento Valley (California), while the signals were less obvious in the other two regions (Columbia River Basin, Washington and Colorado).



### 1.3 Objective of this study

Despite the large number of studies using remote sensing approaches to map irrigated area and irrigation water requirements at various spatial and temporal scales, none of these approaches has attempted to derive actual irrigation water use. To bridge this gap, we propose a new method for estimating IWU from a combination of remotely sensed and modelled reanalysis soil

moisture data. The approach is based on the hypothesis that neither the structure nor the forcing of the model data accounts for artificial water supply, while the microwave soil moisture retrievals do (Kumar et al., 2015; Escorihuela and Quintana-Seguí, 2016). The method is implemented over the contiguous United States by using three state-of-the-art microwave soil moisture products (i.e. based on SMAP, AMSR2 and ASCAT) in combination with MERRA-2 reanalysis soil moisture.

The paper is organised as follows: section 2 provides a general overview of the irrigation landscape in the CONUS. Section 3 covers the utilised satellite-, model- and ancillary data sets and the preprocessing involved. The theoretical and practical aspects of the new methodology to estimate IWU are discussed in section 4. Results are shown and discussed with respect to official reference irrigation data in section 5. Section 6 conclude the study and gives an outlook for follow-on research.

### 2 Study area

#### 2.1 Irrigation practices in the contiguous United States (CONUS)

The amount of water needed by a certain crop for optimal growth mainly depends on three factors: crop type, soil, and climate. Irrigation water need is given by the difference between these requirements for optimal crop growth and effective rainfall. In the largely semi-humid climate of the eastern United States, irrigation is supplemental, which means that irrigation is applied to mostly rain-fed crops during times of insufficient rainfall to achieve higher yields than under rain-fed conditions alone. In

contrast, the predominantly semi-arid climate of the western US makes artificial water supply a necessity, thus requiring full irrigation.

The 2013 Farm and Ranch Irrigation Survey (FRIS) of the National Agricultural Statistics Service (NASS) of the USDA (USDA, 2013) provides selected irrigation data from surveys conducted at approximately 35000 farms using irrigation across

the US. It reports state-level data of both irrigated area and irrigation water withdrawals (IWW) subdivided by specific crop type, water source and irrigation technique. In addition, these estimates are given for crops cultivated outdoors ("in the open"and indoors ("under protection", e.g. horticultural crops grown in greenhouses). Figure 1 shows per state the irrigated area and irrigation water withdrawals limited to crops grown outdoors, as well as irrigation application rates during the 2013 growing season provided by FRIS.

It is likely that the sensitivity of satellite soil moisture retrievals to irrigation increases when the irrigation application efficiency of a particular irrigation system or technique deteriorates. Therefore, we expect higher sensitivity towards gravity-





(e.g. flood and furrow irrigation), and lower sensitivities towards sprinkler- and micro-irrigation systems. Fig. A1 shows a distinct decline in irrigation rates per area from the semi-arid west to the more humid east. The state of Arizona has the highest irrigation rate per area, followed by California and Nevada. Gravity flow systems show the highest rates in California and Arizona, but also depict large values along the Mississippi Delta. This can mainly be attributed to the cultivation of rice which

is primarily grown in these regions and is either flood or furrow irrigated. Finally, micro-irrigation systems are largely limited to the western half of the US.

## 2.2 Focus areas

In addition to the continental-scale analysis, we chose four irrigation hot spots characterised by different climates and irrigation practices within the CONUS (Fig. 2) to comprehensively assess the spatio-temporal dynamics of irrigation. These regions are:

the Sacramento Valley and San Joaquin Valley in the California Central Valley; the Snake River Plain, Idaho; the High Plains, Nebraska; and part of the Mississippi Flood Plain located within the state of Mississippi. For each focus area, we conducted a time series analysis at local scale (Sect. 5.3), as well as a cross comparison with reference data on irrigated area (Sect. 5.5) and irrigation water withdrawals (Sect. A2).

### 2.2.1 Central Valley, California

Traditionally, the California Central Valley accounts for the highest irrigation water withdrawals across the CONUS. Its northern part is characterised by a Mediterranean climate with hot, dry summers, whereas its southern half is defined by both hot and cold semi-arid climates (Kottek et al., 2006). As a result, crop production requires full irrigation. We selected two areas within the Central Valley for the time series analysis: the southern San Joaquin Valley, where several different crop types are cultivated using sprinkler-, furrow- and micro-irrigation-systems, and the northern Sacramento Valley where flood irrigation

for rice is prevalent and which was also investigated by Lawston et al. (2017). Rice production in California is the second largest in the US (NASS, 2012) and relies on large amounts of irrigation water, which is usually supplied by winter snow melt. In the Sacramento Valley, which accounts for 95 % of California's rice yield, rice is typically water-seeded (Linquist et al., 2015). This means that the fields are completely flooded at 10 cm-15 cm depth before planting (usually late April to mid-May), and then seeded with the help of air planes. The fields typically remain flooded throughout the growing season and are only

drained from early September onward, approximately 3 weeks before harvest in September to mid-October.

### 2.2.2 Snake River Plain, Idaho

Idaho accounts for the second largest irrigation water withdrawals in the US after California (NASS, 2012). in Idaho, the Snake River Plain is the most important agricultural area and sprinkler irrigation is the dominant irrigation technique. Similar to the San Joaquin Valley it is characterised by a cold semi-arid climate (Kottek et al., 2006).



### 2.2.3   High Plains, Nebraska

Nebraska is located in the middle of a transitional climate zone which extends longitudinal through the middle of the US. While the climate in western Nebraska is cold semi-arid, the eastern part is humid-continental, characterised by hot summers and year round precipitation (Kottek et al., 2006). For example, irrigation requirements for corn are around approximately $350\,\mathrm{mm}$ in the

west and continuously drop to approximately $150\,\mathrm{mm}$ in the east (reference values obtained from the University of Nebraska-Lincoln). The mainly employed irrigation system is the centre pivot, and the major crops grown are corn and soybean.

### 2.2.4   Mississippi Flood Plain, Mississippi

The Mississippi Flood Plain region is characterised by a fully humid subtropical climate with hot summers (Kottek et al., 2006). Despite the large amounts of rainfall throughout the year, only approximately 30% falls in the summer period when

the major crops are grown (Kebede et al., 2014), thus requiring the use of supplemental irrigation. The dominant crop types include soybean, corn, cotton and rice. Within the Mississippi Flood Plain we chose an area in the state of Mississippi for a local analysis. Here, reports on irrigation water withdrawals (2009 and 2011 growing seasons) are available from the Yazoo Mississippi Delta Joint Water Management District. For the 2011 growing season, average application rates of approximately 180, 400, 330 and $970\,\mathrm{mm}$ are reported for cotton, corn, soybeans and rice respectively, leading to an average of around

$490\,\mathrm{mm}$.

## 3   Data sources

The data sources used in this study are summarised in table 1.

### 3.1   Remotely Sensed Soil Moisture

### 3.1.1   SMAP

The Soil Moisture Active Passive (SMAP) mission was launched in January 2015 and is the second mission exclusively designed for the retrieval of soil moisture together with freeze-/thaw status (Entekhabi et al., 2010). After failure of its radar in July 2015, the radiometer continues to provide measurements in L-band $(1.4\,\mathrm{GHz})$ at a spatial resolution of approximately $40\,\mathrm{km}$. Validation studies have shown that the radiometer meets the target retrieval accuracy of $0.04\,\mathrm{m^3 m^{-3}}$ (ubRMSE) over non-frozen land surfaces free of excessive snow, ice, mountainous terrain, and dense vegetation coverage (Colliander et al.,

2017). In general, L-band is expected to be most suitable for soil moisture retrieval, because it is less affected by vegetation and representative of a deeper soil layer than higher frequency C- or X-band retrievals (Entekhabi et al., 2010). SMAP obatians global coverage every 2-3 days and equatorial crossing times are 06:00 and 18:00 local solar time (LST) for the descending and ascending orbits, respectively. We used both ascending and descending orbit data covering the period of April 2015 to December 2016. We used the passive SMAP_L3_v4 product, which is sampled at $36\,\mathrm{km}$ resolution. In case of overlapping

orbits, we only used the descending (06:00 LST) overpass.



### 3.1.2   AMSR2

AMSR2 is a microwave radiometer on board the GCOM-W1 satellite and provides measurements at $6.9\,\text{GHz}$ (C-band) and three higher frequencies up to $36.5\,\text{GHz}$ (Ka-band) since July 2012 (Imaoka et al., 2010). Daily ascending and descending overpasses are at 13:30 LST and 01:30 LST respectively, achieving global coverage with a spatial resolution of about $40\,\text{km}$

every 1-2 days. The VUA-NASA product used in this study is based on the Land Parameter Retrieval Model (LPRM v6) algorithm, which simultaneously retrieves volumetric soil moisture and vegetation optical depth (VOD) from the observed brightness temperatures (Owe et al., 2008; der Schalie et al., 2016). LPRM is based on a simple radiative transfer equation and partitions the observation into its emission components from soil and vegetation based on the horizontal and vertical polarised brightness temperatures. The data was masked for high VOD values and radio frequency interference (RFI), spatially resampled

to a regular $0.25°$ grid and temporally centred at 12:00 UTC.

### 3.1.3   ASCAT

The Advanced Scatterometer (ASCAT) on board the Meteorological Operational (METOP-A and -B) satellites is operational since October 2006 and is a real aperture radar instrument operating at C-band ($5.255\,\text{GHz}$) and VV-polarization. Local equatorial crossing times are at 21:30 for the ascending overpass and at 09:30 for the descending overpass and global coverage

is achieved every 1-3 days depending on latitude. The TU Wien change detection algorithm (Wagner et al., 1999; Naeimi et al., 2009) is applied to the backscatter coefficients to create time series of relative surface soil moisture for the topmost centimetres of soil. This is accomplished by scaling each observation between reference values representing the historically lowest and highest observed backscatter values, respectively. Soil moisture is provided in degree of saturation (%), and ranges between $0\,\%$ (wilting point) and $100\,\%$ (soil saturation). It has a spatial resolution of $25\,\text{km}$ and is made available on a discrete

global grid (DGG) at a spatial sampling of $12.5\,\text{km}$. In this study, we used a modified version of the EUMETSAT HSAF H111 soil moisture product. The modified version uses a dynamic correction which is expected to better account for inter-annual variability than the original H111 product (Hahn et al., 2017; Vreugdenhil et al., 2016).

### 3.2   MERRA-2 reanalysis soil moisture

The second Modern-Era Retrospective analysis for Research and Applications 2 (MERRA-2) (Bosilovich et al., 2015) is an

atmospheric reanalysis product providing global, hourly fields of land surface and atmospheric conditions for 1980-present at a spatial resolution of $0.625°$ x $0.5°$. It assimilates atmospheric satellite observations using the Goddard Earth Observing System Model (GEOS-5). MERRA-2 uses an observation-based precipitation correction over land to fully correct modelled precipitation at latitudes $<|42.5°|$, with a linear tapering between $|42.5°| <$ latitude $< |62.5°|$ while no correction is applied at more northern and southern latitudes. The precipitation correction has significantly improved the soil moisture simulations

with respect to its predecessor (Reichle et al., 2017a). The soil moisture simulations are representative of the first $10\,\text{cm}$ of soil and are expressed in volumetric units ($[\text{m}^3\,\text{m}^{-3}]$). We explicitly chose MERRA-2 soil moisture in favour of soil moisture from other global reanalysis data sets (e.g. ERA-Interim/Land, GLDAS), because MERRA-2 does not assimilate surface humidity





and surface temperature observations (Reichle et al., 2017b), which are directly impacted by irrigation (Wei et al., 2013; Tuinenburg and Vries, 2017).

### 3.3 Ancillary data

The ESA CCI Land Cover (Bontemps et al., 2013) was used to create a cropland mask for the CONUS. The cropland classes of

the data set represent the period 2008-2012. For a more detailed analysis of the impact of precipitation, we used CPC Unified Gauge-Based Analysis of Daily Precipitation, which covers the CONUS at $0.25°$ native resolution (Chen et al., 2008; Xie et al., 2010). Therefore, it is expected to provide more detailed information than the precipitation data set used to force MERRA-2 SM, which has a $0.5°$ resolution.

### 3.4 Data pre-processing

As data from SMAP were only available from April 2015 onwards, we extended the study period to include available AMSR2 and ASCAT data from 2013 to 2016. Hence, four growing seasons with varying climatic conditions were covered by the AMSR2 and ASCAT sensors, and two growing seasons by SMAP. We assumed a general growing season for the entire CONUS from the start of April to the end of September. All data were spatially matched to a common $0.25°$ regular grid using nearest-neighbour resampling. In order to constrain the analysis to areas where irrigation is feasible, we masked all pixels with $< 5\,\%$ of

fractional cropland area based on ESA CCI land cover for 2010 (Bontemps et al., 2013). Unreliable observations in the satellite data were masked applying their respective quality flags for frozen soil, dense vegetation and radio frequency interference.

The spatial representativeness and observation depth slightly differ among the the various remote sensing products and modelled soil moisture. MERRA-2 SM is simulated for a fixed $10\,\mathrm{cm}$ thick soil layer (Bosilovich et al., 2015) and thus shows

more inertia to changes in the water balance (i.e. through precipitation) than the remotely sensing data. Besides, ASCAT is provided in a different unit than the other products. To account for these systematic differences between products, we applied a linear re-scaling approach (Brocca et al., 2013), which forces the satellite soil moisture time series $\Theta^{sat}$ to have the same mean $\mu$ and standard deviation $\sigma$ as the modelled soil moisture $\Theta^{mod}$:

$$\Theta^{sat}_{rescaled} = \frac{\Theta^{sat} - \mu(\Theta^{sat})}{\sigma(\Theta^{sat})}\sigma(\Theta^{mod}) + \mu(\Theta^{mod}) \tag{1}$$

It is likely that over irrigated areas $\mu(\Theta^{sat})$ increases during the respective irrigation period, which will alter the scaling parameters. We expect that this should not affect the temporal evolution of changes in soil moisture. However, the influence of irrigation on the temporal variability of soil moisture (depending on the type of irrigation, general climate conditions, etc.) is a source of uncertainty. In particular over very dry regions, the model soil moisture may never reach saturation, while the remotely sensed soil moisture data does (due to irrigation). Since the variable of interest is irrigation amount, volumetric soil

moisture in $\mathrm{m^3\,m^{-3}}$ is converted to the corresponding water column depth $D_{watertable}(\mathrm{mm})$ by multiplying it with the depth of soil $D_{soil}$ for which the soil moisture simulations are representative. Thus, for the layer $0-10\,\mathrm{cm}$, e.g., $0.3\,\mathrm{m^3\,m^{-3}}$ correspond to a $30\,\mathrm{mm}$ water column covering the unit area of $1\,\mathrm{m^2}$.




## 4 Methods

### 4.1 Theoretical prove of retrieving irrigation water use from microwave remote sensing

Kumar et al. (2015) first proposed the idea of inferring irrigation from a positive bias between remotely sensed and modelled soil moisture, induced by seasonal water application during the dry season. This ideas is based on two key assumptions: first,

the satellite soil moisture products are sensitive to large scale irrigation (as partly confirmed by Escorihuela and Quintana-Seguí (2016); Lawston et al. (2017)) and second, the model does not account for irrigation, neither explicitly (i.e., in the formulation) nor implicitly through the assimilation of surface humidity or surface temperature observations, which are affected by irrigation (Wei et al., 2013). We build on these assumptions and introduce a new metric to estimate IWU from the difference between satellite-observed and modelled soil moisture. The soil water balance equations describing the respective change in

soil moisture for each time step t [d] are described by

$$\frac{d\Theta^{sat}}{dt} = P(t) + I(t) - ET(t) - R(t) - \Delta S_{rest} \tag{2}$$

for the satellite observations and

$$\frac{d\Theta^{mod}}{dt} = P(t) - ET(t) - R(t) - \Delta S_{rest} \tag{3}$$

for the model simulations. $P[\text{mm}]$ is precipitation, $I[\text{mm}]$ is irrigation, $ET[\text{mm}]$ is evapotranspiration, $\Theta^{sat}$, $\Theta^{mod}[\text{mm}]$ are

satellite and modelled surface soil moisture, respectively, converted to water column depth. $\Delta S_{rest}[\text{m}^3\,\text{m}^{-3}]$ describes water storage changes below the surface layer, including drainage. Subtracting ( 3) from ( 2) yields

$$I(t) = \frac{d\Theta^{sat}}{dt} - \frac{d\Theta^{mod}}{dt} \tag{4}$$

Hence, estimating irrigation from differences between the temporal variations of satellite and model SM is theoretically feasible.

### 4.2 Deriving irrigation water use

We define an irrigation event as a simultaneous increase in satellite soil moisture ($\frac{d\Theta^{sat}}{dt} > 0$) and a decrease or stagnation in model soil moisture ($\frac{d\Theta^{mod}}{dt} \leq 0$). This means that rainfall did not cause the satellite-observed increase in soil moisture, which over agricultural land was very likely a result of irrigation. For each event, the amount of irrigation water leading to the increase is derived as the difference $\frac{d\Theta^{sat}}{dt} - \frac{d\Theta^{mod}}{dt}$, if the change in satellite is significant (i.e. above the noise level). The latter

is accounted for by applying a threshold of relative soil moisture change $thresh_\Theta$ (see section 4.3 and appendix A). We then calculate seasonal irrigation water use ($IWU$) summing up the approximated difference quotients over the growing season period:

$$IWU = \int_{i_{SOS}}^{i_{EOS}} \left( d\Theta_i^{sat} - d\Theta_i^{mod} \right) dt \approx \sum_{i=SOS}^{EOS} \Delta\Theta_i^{sat-mod} \tag{5}$$





where

$$\Delta\Theta_i^{sat-mod} = \begin{cases} \Delta\Theta_i^{sat} - \Delta\Theta_i^{mod}, & \text{if } \Delta\Theta_i^{sat} \geq \Theta_{thresh} \text{ and} \\ 0, & \text{otherwise} \end{cases}$$

with

$$\Delta\Theta_i^{sat} = \Theta_i^{sat} - \Theta_{i-n}^{sat},$$

$\Delta\Theta_i^{mod} = \Theta_i^{mod} - \Theta_{i-n}^{mod}$

$IWU$ is the accumulated irrigation water use from the start ($i_{SOS}$) until the end of the growing season ($i_{EOS}$). According to the crop calendars provided by Portmann et al. (2010) and the USDA Planting and harvesting dates handbook (NASS, 2010) the period 1 April- 30 September generally covers the growing season of most crops receiving irrigation water in the CONUS. $\Theta_i^{sat}$ and $\Theta_i^{mod}$ are satellite and model SM at day $i$, $thresh_\Theta$ denotes the relative soil mosisture threshold and $\Theta_{i-n}^{sat}$ and $\Theta_{i-n}^{mod}$ are the

last available soil mosisture observations with a data gap of $n$ days. If an irrigation event is detected during an observation gap of $> 4$ days, we check if there has been a significant increase in the model soil moisture (e.g. due to rainfall) within that period. When more than one significantly positive model slopes (or precipitation events) occur during the gap-period, we cannot say for sure if the observed increase in soil moisture was due to irrigation or precipitation and therefore conservatively disregard the potential irrigation event.

### 4.3  Masking spurious irrigation detections

It is essential to differentiate between irrigation signals and high frequency noise in the satellite data. For this purpose, we apply a threshold $thresh_\Theta$ to the relative changes in satellite soil moisture similar to the approach applied by Dorigo et al. (2013) to detect spurious in-situ data:

$$\frac{\Theta_t^{sat} - \Theta_{t-n}^{sat}}{\Theta_{t-n}^{sat}} \geq 0.12 \equiv thresh_\Theta \tag{6}$$

Based on a sensitivity analysis (Sect. A) we concluded that a threshold of $thresh_\Theta \equiv 0.12$ is an appropriate generic choice for the whole CONUS.

Potential errors may arise when the model forcing misses or creates false rainfall events. In addition, because of differences in timing of the estimates and differences in represented soil depth between remotely sensed and modelled soil moisture, their

response to precipitation events may differ as well. This can lead to spurious irrigation events when irrigation is estimated at days with rainfall. We therefore used precipitation from MERRA-2 and CPC precipitation data to double check if the observations and/or model estimates are affected by rainfall and removed them from the analysis if they were. Furthermore, if a potential irrigation signal coincides with preceding rainfall we assume that irrigation is unlikely and disregard the event.



In some extreme cases, capillary rise from deeper soil layers or run-on can wet the top soil. Theoretically, these conditions are reflected by the satellite soil moisture retrievals, but absent in the model soil moisture simulations (i.e. if such effects are not accounted for in the soil hydrology formulation of the LSM (McColl et al., 2017)). However, at the large spatial scales represented by the employed satellite (approximately $25\,\mathrm{km}$) and model soil moisture products (approximately $50\,\mathrm{km}$), very

few pixels are expected to show positive $\Delta\Theta_i^{sat}$ or $\Delta\Theta_i^{mod}$ in the absence of precipitation or irrigation.

## 5   Results and Discussion

### 5.1   Growing season correlations between satellite and model soil moisture

To investigate the potential detectability of IWU, we investigated the correlation between satellite and modelled soil moisture during the growing season ((Figure 3). We computed the correlation separately for dry (precipitation$=0$; $\overline{r}_{dry}$) and wet con-

ditions (precipitation$>0$; $\overline{r}_{wet}$). If $\overline{r}_{dry}$ is low or negative over agricultural areas which are known to be irrigated (as inferred from the MIrAD-US product) while $\overline{r}_{wet}$ is strongly positive, this is a strong indication of irrigation.

Over non-agricultural land cover, low growing season correlations between SMAP and MERRA-2 soil moisture are observed over the densely vegetated south- and north-eastern US, and over parts of the arid south-west (Figs. 3a, 3b). AMSR2

exhibits low correlations in coastal areas, complex terrain and over dense vegetation cover (Fig. 3c, 3d). ASCAT shows negative correlations against MERRA-2 over the arid south-western deserts and the densely vegetated coastal north-west and south-east (Fig. 3e, 3f). Overall, for each satellite-model pair there is a clear reduction of $\overline{r}_{dry}$ with respect to $\overline{r}_{wet}$ over several irrigation hot spots within the CONUS.

### 5.1.1   Central Valley

Over the Central Valley, SMAP shows moderate to high $\overline{r}_{wet}$ with MERRA-2, except for the Sacramento Valley in northern California. In contrast, $\overline{r}_{dry}$ is moderately to strongly negative over the southern San Joaquin Valley, which indicates that an irrigation signal is indeed observed by the satellite sensor. The fact that $\overline{r}_{dry}$ is comparable, if not higher than $\overline{r}_{wet}$ over the Sacramento Valley should be attributed to the special characteristics of the prevalent rice irrigation. In the Sacramento Valley, a permanent flood of 10-15cm is usually maintained during the whole growing season before fields are drained in preparation

for harvest (Linquist et al., 2015). Hence, irrigation water remains observable during both wet- and dry periods of the growing season, and the impact of irrigation on $\overline{r}$ actually increases for the wet period with respect to the dry period. In contrast, ASCAT exhibits high correlations with MERRA-2 in this region. This may be due to specular reflection of the radar signal from the rice, which would cause a signal that looks similar to one coming from a dry soil. Both SMAP and ASCAT show moderate to high negative correlations against MERRA-2 in the heavily irrigated San Joaquin Valley. Concerning AMSR2 SM there is no

clear pattern of discrepancy between $\overline{r}_{wet}$ and $\overline{r}_{dry}$ in the Central Valley.



### 5.1.2 Snake River Plain

Over the Snake River Plain, ASCAT has a clear signal that could be attributed to irrigation. Particularly in the central and western-most areas along the Snake River, $\overline{r}_{dry}$ depicts a strong negative correlation with MERRA-2, while $\overline{r}_{wet}$ is moderately positive. Moderately negative $\overline{r}_{dry}$ obtained for SMAP shows a good alignment with areas known to be irrigated in the Snake

River Plain. Although the correlation is less negative than for ASCAT, the spatial pattern is resembled more clearly. Here, AMSR2 shows slightly more negative $\overline{r}_{dry}$ than $\overline{r}_{wet}$ over agricultural land cover, but the spatial pattern appears to be less reliable than for the other satellite products.

### 5.1.3 High Plains

The ASCAT product is the only one to show a distinct pattern of negative correlation over the irrigated part of the Nebraska

Plains. While $\overline{r}_{wet}$ shows weak positive correlations, $\overline{r}_{dry}$ reveals strong negative $\overline{r}$, suggesting that an irrigation signal is entailed in the ASCAT signal. However, this pattern cannot be reliably attributed to irrigation practices as ASCAT shows low correlations over the entire Corn Belt region, where agriculture is generally known to be rain-fed (see Fig. 2). Vegetation scattering effects from the corn canopies are a plausible explanation for the observed deviation. As the corn plants reach their maximum height (up to approximately 3m) towards the end of the growing season, the C-band backscatter signal will

increasingly be composed of canopy backscatter and canopy-soil double bounce reflections, while sensitivity to actual soil moisture decreases (Daughtry et al., 1991; Joseph et al., 2010).

### 5.1.4 Mississippi Flood Plain

Lastly, all products show low $\overline{r}$ values over the Mississippi Flood Plain, although with varying magnitudes. In this region, ASCAT shows the lowest negative correlation, followed by AMSR2 and SMAP. Moreover, for all SM products $\overline{r}_{dry}$ is lower

than $\overline{r}_{wet}$.

### 5.1.5 Other regions

Figure 3 also reveals that strong negative $\overline{r}_{dry}$ (in combination with moderately high $\overline{r}_{wet}$) based on ASCAT align well with areas known to be irrigated over the Columbia River Basin, Washington. Correlations based on SMAP loosely agree with this pattern and AMSR2 only has few valid observations over this region. In addition, both ASCAT and SMAP have patterns of

$\overline{r}_{dry} < \overline{r}_{wet}$ over an irrigated region in south-western Georgia. In contrast, AMSR2 shows moderately high positive $\overline{r}$ over this region.

To determine the sensitivity of the growing season correlation $\overline{r}$ between satellite and model soil moisture to variations in fractional irrigated area within a pixel, we examined their relationship with irrigation intensities derived from the MIrAD-US

irrigated area data set (Pervez and Brown, 2010). However, no evidence of a negative linear relationship between the two variables was found (not shown), as at low irrigation fractions $\overline{r}$ is mostly dominated by effects originating from the remaining



land cover types. Overall, the results obtained by separately analysing the spatial patterns of $\overline{r}_{dry}$ and $\overline{r}_{wet}$ between satellite and model soil moisture largely support the hypothesis that over areas known to be irrigated, the remotely sensed soil moisture signal deviates from modelled soil moisture, given that the model does not explicitly account for irrigation (which is the case for MERRA-2). Hence, the overall hypothesis of this study, which is that IWU can be inferred from differences between the

temporal variations of the remotely sensed and modelled soil moisture is corroborated.

## 5.2  Spatial patterns of estimated irrigation water use

Spatial plots of mean annual estimated irrigation water use $\overline{IWU}$ (i.e. averaged over the study period of 2013-2016) (Figs. 4a-4c) suggest that all satellite products are able to identify the extensive irrigation applied in the California Central Valley. Here, SMAP derived $\overline{IWU}$ clearly resembles the irrigation patterns of the MIrAD-US data set in the northern Sacramento Val-

ley and southern San Joaquin Valley (Fig. 2). AMSR2 and ASCAT derived $\overline{IWU}$ is generally higher than SMAP and extends throughout the whole California Central Valley. Although small in magnitude, the $\overline{IWU}$ pattern derived from ASCAT over the central Snake River Plain is spatially distinct. Similarly, AMSR2 shows a clear signal over the western to central Snake River Plains. Concerning the Nebraska Plains, only AMSR2 $\overline{IWU}$ shows patterns that agree with the the MIrAD-US. Over the Mississippi Flood Plain, ASCAT shows the highest $\overline{IWU}$, followed by AMSR2.

ASCAT-derived $\overline{IWU}$ seems to be affected by vegetation effects in the Corn Belt region and in the south-eastern US. For all sensors, the method fails to detect $\overline{IWU}$ in many irrigated areas, especially those along the High-Plains-Aquifer (Nebraska, Kansas, Texas), which extends from the northern to the southern central US. A plausible explanation for missing these areas is that many farmers in these regions practice supplemental irrigation, thus resulting in a less distinguishable irrigation signal.

In addition, the centre pivot irrigation systems, which are mainly used in this region, have much higher water application efficiencies compared to the flood- and furrow irrigation systems used in the Sacramento Valley and Mississippi Flood Plain (see Fig. A1). Furthermore, the global relative threshold of $thresh_{\Theta} = 12\%$ also masks irrigation signals in regions such as the Columbia River Basin, Washington (results not shown). Thus, a more site-specific threshold at each pixel might lead to improved detectability.

## 5.3  Temporal behaviour of soil moisture and IWU and in the four focus regions

For a more detailed analysis on the impact of climate, crop type, and irrigation practice on the method performance, we compared remotely sensed and modelled soil moisture time series, and monthly $IWU$ estimates at an irrigated (green crosses in Fig. 2) and a non-irrigated pixel (orange crosses in Fig. 2) in the four focus areas (Fig. 5).

### 5.3.1  Central Valley

At the irrigated pixel in the San Joaquin Valley, the impact of irrigation on the remotely sensed soil moisture signal is evident (top panel in Fig. 5a). While MERRA-2 soil moisture decline in the irrigated and non-irrigated pixels reflects the absence



of precipitation, typically from mid-May until mid-October, ASCAT and SMAP soil moisture in the irrigated pixel start to increase in June until reaching their maximum in July-August and gradually declining towards the end of the growing season. In contrast, remotely sensed soil moisture in the adjacent non-irrigated pixel (bottom panel) remains close to zero throughout the growing season. The temporal behaviour of SMAP at the irrigated pixel location was extensively evaluated by Lawston

et al. (2017), who showed that SMAP soil moisture correctly reflects the onset of flood irrigation, the dry down associated with plants breaking through the water surface (which attenuates the SM signal), and lastly field drainage. Even though Lawston et al. (2017) used the enhanced 9 km sampling SMAP product, we find very comparable temporal characteristics for the native 36 km resolution product (Fig. 5b). ASCAT soil moisture seems to be impacted by specular reflection of the active radar signal from the flooded rice fields, leading to very low backscatter and, hence, soil moisture values. As a result, particularly during

the 2013 growing season ASCAT soil moisture remains at- or very close to its minimum early in the growing season. ASCAT soil moisture starts to increase in early to mid-July when the rice starts to break out of the water. Initially, the increase is primarily the result of double-bounce effects from the rice canopies, while at later growth stages this turns into volume scattering (Nguyen et al., 2015). Of the three products, ASCAT soil moisture is the last to reach its growing season maximum between mid- to late August, followed by a decrease throughout September. At the irrigated pixel, AMSR2 soil moisture shows large

fluctuations during the growing season, while at the non-irrigated pixel it has few valid observations, which makes it difficult to compare both pixels. AMSR2 soil moisture shows similar characteristics with respect to SMAP and is able to sense the onset of flood irrigation, but reaches saturation a few weeks earlier and already starts to dry down before SMAP reaches its soil moisture maximum. Moreover, after reaching a minimum in late July to early August, AMSR2 soil moisture starts to increase again.

Comparing the estimated monthly IWU (bottom sub-panels) at the adjacent irrigated and non-irrigated pixels suggests that the method is skillful in detecting irrigation from all considered sensors, especially during the comparatively dry years of 2013 and 2014, when a prolonged drought affected the state of California. AMSR2 provides highest IWU estimates, possibly due to the high noise levels in the soil moisture data. In general, a spurious irrigation signal remains at the non-irrigated pixel, which may be due to noise in the satellite soil moisture retrievals. At the non-irrigated pixel, ASCAT- and AMSR2-based

IWU retrievals seem to be more affected by noise than SMAP. The 2015 growing season was unusually wet, which at the non-irrigated pixel in the spurious detection of irrigation for all satellite products. Generally, especially SMAP and ASCAT products are skillful in detecting the seasonality of irrigation over the San Joaquin Valley.

### 5.3.2   Snake River Plain

At Snake River Plain, all satellite soil moisture products show a clear irrigation signal at the irrigated pixel (Fig. 5c), which is
not visible in the non-irrigated pixel. Consequently, considerable IWU is estimated for the irrigated pixel, while for ASCAT and SMAP the estimated IWU at the non-irrigated pixel is close to zero. AMSR2 soil moisture retrievals are noisier, which results in the detection of some spurious irrigation at the non-irrigated pixel, although significantly smaller than at the irrigated pixel. We argue that the higher spatial sampling of the employed ASCAT data is advantageous for IWU estimation, as the irrigated area within the Snake River Plain is quite narrow.



### 5.3.3 High Plains

ASCAT soil moisture content is higher than MERRA-2 during the drier periods of the growing season, indicating sensitivity to the typically employed supplemental irrigation (Fig. 5d). However, the relative changes in ASCAT soil moisture are $< 12\%$ and therefore do not qualify as rigorous irrigation events based on our methodology. AMSR2 provides the largest derived IWU at the irrigated pixel, but is also affected by noise at the non-irrigated pixel. In this area, the influence of irrigation on the remotely sensed soil moisture signal is much more subtle, if significant at all. This can be attributed to two factors: First, due the abundant rainfall during the growing season only supplemental irrigation is applied in this area. Second, center pivot irrigation systems usually have much higher application efficiencies $(75 - 95\%)$ than gravity irrigation systems $(40 - 80\%)$. Therefore, less water needs to be applied to achieve comparable plant growth, rendering a less distinct irrigation signal in the soil moisture product.

### 5.3.4 Mississippi Flood Plain

The difference in soil moisture behaviour between the irrigated and non-irrigated pixel is much more pronounced for AMSR2 than for the other satellite products (Fig. 5e). This is also reflected by AMSR2-derived IWU at the irrigated pixel, which agrees well with the expected seasonality of irrigation, which peaks in August. At the same time AMSR2-based IWU estimates are close to zero at the non-irrigated pixel. ASCAT soil moisture shows a similar seasonality at the irrigated pixel, but is more affected by noise at the non-irrigated pixel. SMAP soil moisture sustains saturation throughout the first half of the growing season, which could be either caused by flood irrigation for rice, or point at a problem in the soil moisture retrieval algorithm. At least for AMSR2, IWU shows a meaningful derived seasonality.

### 5.4 Evaluation of estimated irrigation water use against state-level reference water withdrawals

We evaluated the agreement between mean IWU $\overline{IWU}$, aggregated for each satellite-model pair to the state level, and reported irrigation water withdrawals from the 2013 FRIS (USDA, 2013)($IWW_{FRI}$). The median correlation $R$ values for SMAP-, AMSR2- and ASCAT-based $\overline{IWU}$ and $IWW_{FRIS}$ are 0.79, 0.56, and 0.36, respectively. For all satellite datasets, California is correctly identified as the largest consumer of irrigation water, which indicates the overall potential of coarse resolution microwave soil moisture data in estimating $IWU$. However, the root-mean-squared-difference (RMSD) and bias between observed $\overline{IWU}$ and $IWW_{FRIS}$ indicate a clear underestimation. The lowest RMSD of 5.01 $km^3$ was found for AMSR2, but values for SMAP and ASCAT are very similar. ASCAT has the lowest bias $(-2.06\ km^3)$, but the bias based on the other products are similar. We further discuss the potential reasons for the generally large biases observed in section 5.6. On average, $\overline{IWU}$ based on SMAP provides the closest similarity with $IWW_{FRI}$).

### 5.5 Evaluating irrigated area estimates with the MIrAD-US data set

We compared spatial patterns of total average $\overline{IWU}$ estimates with the MIrAD-US data set at 0.25 resolution. To be able to compute a confusion matrix, we converted the continuous ranges of the two data sets into binary representations of irrigated

(c) Author(s) 2018. CC BY 4.0 License.





areas. For MIrAD-US, this was accomplished by labelling only areas with $>= 5\%$ irrigation as irrigated. Estimated $\overline{IWU}$ was converted in a similar way by considering only pixels where $\overline{IWU} \geq 20mm$ as irrigated. To comparing irrigation estimated from each satellite-model pair with MirAD-US, we computed the error of omission, the error of commission, the overall accuracy, and Cohen's kappa $\kappa$, which is a measure of how the classification results compare to values assigned by chance (Table 2).

In California, irrigated area estimates based on $\overline{IWU}$ shows very good agreement with MIrAD-US. SMAP-based irrigated areas provides the highest scores for all metrics: the overall accuracy (OA) is 77.68%. A commission error (EoC) of 21.43% in combination with an omission error (EoO) of 4.94% indicates that we somewhat overestimate the reference and a kappa score of $\kappa = 0.33$ illustrates a fair agreement. ASCAT has a similar performance, but shows a slightly higher overestimation

($EoC = 24.51\%$), thus resulting in a lower overall accuracy and $\kappa$ score. In California, AMSR2 performs worst, but still shows an acceptable overall accuracy of 68.75%. In Idaho, of all satellite products AMSR2 clearly shows the highest agreement with the reference ($OA = 59.02\%$), but $EoC$ and $EoO$ are equally high at approximately 41%, indicating a moderate amount of confusion in the classification. As a result of a strong under-classification, in Nebraska there is hardly any agreement between IWU-based and MIrAD-US irrigated area. Over the Mississippi Flood Plain, AMSR2-based IWU shows good (among the

products by far the best) agreement with the reference data ($OA = 71.11\%$, $\kappa = 0.32$). We argue that, due to the previously observed problems regarding the representation of soil saturation, both ASCAT and SMAP soil moisture are unreliable in this region. ASCAT-based IWU depicts a high over-estimation ($EoC = 62.65\%$) while SMAP-based IWU does not classify any irrigated areas at all ($EoO = 100\%$). For CONUS as a whole, SMAP depicts the best agreement with MIrAD-US irrigated area, which is reflected by an overall accuracy of 74.03%. However, SMAP fails to correctly classify approximately 90% of

areas irrigated according to the MIrAD-US. AMSR2 shows the second best agreement ($OA = 66.82\%$) and misses fewer pixels than SMAP ($EoO = 72.7\%$), but in contrast shows a higher over-estimation ($EoC = 67.64\%$). ASCAT shows the lowest agreement with an OA of 61.88%.

The results obtained for California are encouraging and emphasise the potential of coarse scale microwave soil moisture

retrievals in correctly detecting the spatial patterns of irrigation. However, consistent with the findings of estimated irrigation volumes, irrigated area estimates reflect a general pattern of underestimation with respect to the MIrAD-US data set. The results further indicate that in areas such as Nebraska, where the climate is semi-humid in large parts of the state and irrigation is mostly supplementary, the method fails in detecting the irrigation signal.

### 5.6 Sensitivity of microwave soil moisture products to irrigation

By qualitatively examining the obtained results, we find that the sensitivity of the employed microwave soil moisture retrievals to irrigation and the performance with respect to reported irrigation data particularly depend on the following factors:

1. **Spatial resolution of the microwave soil moisture products and topography**

   Likely, the largest restriction is the coarse scale of the satellite soil moisture retrievals with respect to the average field



size. For instance, the area irrigated by a typical centre pivot system (i.e. $500\,\mathrm{m}$) is approximately $50\,\mathrm{ha}$, which only accounts for approximately $0.0003\,\%$) of the satellite footprint area. Thus, around 3200 centre pivot systems are needed to create a uniformly irrigated area covering the remotely sensed footprint. In the CONUS, areas with large irrigation fractions exist in the eastern half of the country, but irrigation in the arid western half is a lot more heterogeneous. In

these areas, irrigation usually mainly depends on surface water supply and is therefore reserved to narrow river valleys such as the Colorado River valley. As a consequence, coarse scale microwave soil moisture products may be insensitive to locally significant (but insignificant with respect to the scale of the satellite footprint) irrigation due to the small scale of the irrigation practices and surrounding complex topography (e.g. mountains, valley-transitions, water bodies). However, as depicted by figure A1 and the latest Agricultural Census from USDA, irrigation water application rates are

the highest in arid climates. We therefore expect that these drawbacks significantly contribute to the underestimation of reported irrigation water withdrawals.

2. **Climate**

As discussed in section A, the method in its current formulation is only applied to rain-free periods during the growing season. We believe that this constraint accounts for a substantial part of the underestimation of $\overline{IWU}$ with respect

to reported $IWW$. If rainfall cannot meet the plant's total daily evaporative demand, farmers may decide to irrigate even at rainy days. Actual, farmers often irrigate at days with rainfall, at which evapotranspiration rates are lower and, hence, irrigation water loss decreases. Nevertheless, we did not come up with an adequate way of decomposing the impact of rainfall and irrigation in the soil moisture signal on a daily basis. At daily temporal sampling, in some cases satellite and model soil moisture show markedly different responses to precipitation events both in terms of temporal

characteristics and intensity. This results in spurious irrigation events, which motivated us to constrain the method to dry periods. When full irrigation is applied in arid climates, the microwave soil moisture retrievals generally show promising skills in detecting the irrigation signals. In contrast, for the predominantly semi-humid climate (e.g. High Plains and Mississippi Flood Plain) irrigation mainly aims at increasing yield or bridging dry periods (i.e. supplementary irrigation). Consequently, less irrigation water is applied and thus the microwave soil moisture retrievals may not appropriately

capture less pronounced soil wetting.

3. **Crop type and irrigation system**

Water requirements naturally vary between crop types. Of the main crop types grown in the CONUS, alfalfa and rice typically require most irrigation water. Particularly flood irrigation for rice leads to a strong irrigation signal in the Sacramento Valley. The signal of flood irrigation is less distinct in the semi-humid climate of the Mississippi Flood Plain, yet

SMAP sensed a prolonged period of soil saturation which may be attributed to flooded fields (section 5.3, Fig. 5e). However, the consistent rainfall in this region reduces the method performance, as the irrigation and precipitation signals in the soil moisture data cannot be disentangled. On the other hand, the extensive supplementary irrigation applied by centre pivot systems in the High Plains does result in a minor irrigation signal in the soil moisture retrievals (section 5.3, Fig. 5d). Intriguingly, the same dynamic was observed for other irrigated areas along the Ogallala Aquifer (Kansas, Ok-



lahoma, Texas; not shown).

We suggest that the differences in detectability may be related to the application efficiency of a particular irrigation
system, which is defined as the ratio between the average low quarter depth of water added to root zone storage and
the average depth of water applied to the field (in mm) (Pereira et al., 2002). Gravity irrigation systems have the lowest
efficiency (approximately 60 %), followed by sprinkler- (approximately 75 %), and micro-irrigation systems (approx-
imately 90 %). Consequently, microwave soil moisture retrievals are expected to be most sensitive to flood irrigation,
followed by sprinkler- and micro-irrigation. The highest irrigation water consumption per area occurs in the arid west
and particularly in the south-west (Fig. A1. Besides, each farmer separately controls his fields and focuses on a different
variety of crops based on market conditions, which in turn require different irrigation amounts and timing. This means
that between satellite overpasses only a fraction of the fields within the satellite footprint will have received irrigation
(based on the individual water management of each farmer). In addition, a centre pivot irrigation system takes between
12-24 hours to complete a full rotation circle, which means that at the time of satellite overpass only a fraction of a field
has recently received irrigation. The same is true for other irrigation techniques, as irrigation equipment (e.g. pipes) has
to be manually transferred across fields.

4. **Satellite observation system and wavelength**

Our results indicate that the observation system (i.e. active or passive remote sensing system) have an impact on the
sensitivity of soil moisture retrievals to irrigation. For instance, the water applied by certain types of irrigation (i.e. flood
irrigation for rice) could not be comprehensively detected by the active ASCAT sensor due to specular reflection of the
radar signal (see section 5.3, Fig. 5b). On the other hand, the active microwave data provided valuable information on the
timing of both flood irrigation onset and field drainage and additionally allowed insights into the crop development cycle
(i.e. backscatter increases when the vegetation starts to break through the standing water surface, potentially causing
double bounce effects (Nguyen et al., 2015)). These dynamics largely agree with the observations reported by Lawston
et al. (2017). Regarding the observation wavelength, we found that the SMAP L-Band data showed more sensitivity than
AMSR2 C-band to the flood maintenance flow that is commonly established after the start of the growing season at the
rice-irrigated site in the Sacramento Valley, California. However, we cannot safely conclude if this is actually due to the
observation wavelength or to differences in the retrieval algorithms.

## 6   Conclusions and Outlook

In this paper we presented a new methodology to derive irrigation water use at monthly time scales by combining microwave
remote sensing and modelled soil moisture data. We first assessed if irrigation impacts the correlation between remotely sensed
and modelled soil moisture and found that the growing season correlations between each satellite-model pair (SMAP, AMSR2,
and ASCAT against MERRA-2) are significantly lower over major irrigation areas throughout the CONUS. Hence, deriv-
ing IWU from differences between satellite and model data is theoretically possible. We then derived IWU estimates over



the CONUS for the period 2013-2016 and evaluated our estimates, aggregated per state, with reports on state-level irrigation water withdrawals from the 2013 Farm and Ranch Irrigation Survey (USDA, 2013). Of all satellite products, SMAP-derived IWU showed the highest correlation between state-aggregated observed and reported irrigation volumes (r=0.79), followed by AMSR2 (r=0.56) and ASCAT (0.36). Moreover, we compared the spatial IWU patterns against the MIrAD-US dataset (Pervez and Brown, 2010). Again, SMAP-derived IWU patterns showed highest agreement with the MIrAD-US, followed by AMSR2 and ASCAT.

However, for all satellite products, derived IWU is significantly lower than reported irrigation water withdrawals. In line with previous studies (Escorihuela and Quintana-Seguí, 2016), we argue that this discrepancy can be mainly attributed to the coarse resolution of the satellite soil moisture retrievals, which in many regions does not allow for resolving the irrigation signal at the field scale, or for areas of small scale irrigation. Besides, the derived IWU relies on the quality of the soil moisture observations, which are impacted by topography, vegetation effects, instrument noise, and the observation principle itself (active versus passive microwave observations). Furthermore, the ability to extract IWU is controlled by the sensitivity of the overall soil moisture signal to irrigation, which is driven by the type and frequency of irrigation, its timing with respect to the satellite overpass, and climate. For example, our method failed to detect IWU in areas with humid growing seasons where irrigation is only supplemental. Despite these major drawbacks, we found that the seasonality of observed irrigation water use is meaningful over several irrigation hot spots such as the California Central Valley, the Snake River Plain, and the Mississippi Flood Plain.

Many of the current limitations can be overcome by using imagery of higher spatial resolution (providing improved capacity to resolve the local irrigation signal within the satellite footprint area) and temporal resolution (providing observations closer to the actual irrigation time). Suitable candidates are e.g., the spatially interpolated SMAP enhanced 9 km product (Chan et al., 2018), the SCATSAR-SWI product with a 1 km spatial and daily temporal resolution obtained from the fusion of active microwave remote sensing 25 km Metop ASCAT soil moisture (Bauer-Marschallinger et al., 2018), or soil moisture products at approximately 1 km based on Cyclone Global Navigation Satellite System (CYGNSS) (Chew and Small, 2018). Also multi-satellite products, such as the ESA Climate Change Initiative Soil Moisture product (Dorigo et al., 2017; Liu et al., 2012) offer a great potential to increase spatial and temporal resolutions, provided that the original soil moisture variations are maintained in the merged product.

Despite the current limitations observed, our findings highlight the potential of using microwave soil moisture retrievals for estimating intra- and inter-annual variations in actual IWU and indicate the overall usefulness of the proposed method. IWU estimates based on microwave soil moisture observations can provide both stand-alone information and synergistic value in combination with methods commonly used to estimate irrigated area or potential evaporative demand from optical or thermal data. Based on past and current microwave satellite missions, remotely sensed soil moisture has the potential to provide information on irrigation water use over the last four decades, which can be used to force climate models and assess the impact of irrigation on regional climate.





**Appendix A: Optimization of the noise threshold $thresh_{\Theta}$**

**A1    Optimization based on soil moisture time series in the four focus-regions**

We apply a minimum threshold $thresh_{\Theta}$ to separate increases in soil moisture stemming from true irrigation from disturbing impacts like dataset noise. We optimised $thresh_{\Theta}$ by maximising the relative difference between IWU estimated at an irrigated

($PI$) and a non-irrigated pixel ($PNI$) in each focus area (Sect. 2.2) and for each satellite product. This is done through the following function:

$$\frac{\overline{IWU}_{PNI} - \overline{IWU}_{PI}}{\overline{IWU}_{PI}} \to min. \tag{A1}$$

where $\overline{IWU}_{PI}$ and $\overline{IWU}_{PNI}$ are mean annual IWU estimated at $PI$ and $PNI$ respectively (Fig. A2). Even though there is no single $thresh_{\Theta}$ that leads to a minimum for all sensors and all focus areas, we find an overall $thresh_{\Theta}$ of 0.12 . Moreover, the

data clearly show that SMAP soil moisture has a lower noise level (no data points mean that $\overline{IWU}_{PNI}$ is zero) in comparison with AMSR2 and ASCAT soil moisture across the different climatic conditions and irrigation practices reflected by the four focus regions. The reason why the estimated IWU at the $PNIs$ is non-zero is two-fold: First, the true spatial resolution of the satellite SM products (approximately $40\,km$ for SMAP and AMSR2, approximately $25\,km$ for ASCAT) is coarser than the spatial sampling of the common data grid ($25\,km$) and thus, if the choice of $PNI$ is very close to $PI$, the soil moisture retrieval

at $PNI$ may as well be affected by irrigation. Second, noise or deficiencies in the soil moisture retrieval can result in spurious irrigation signals.

**A2    Optimization based on the correlation of estimated irrigation water use with reported water withdrawals**

To supplement the local optimization analysis, we examined how different noise $thresh_{\Theta}$ values affect the agreement between observed $IWU$ and reported irrigation water withdrawals $IWW_{FRIS}$. For this purpose, annual mean irrigation water use

$\overline{IWU}$ derived from each satellite-model pair by applying different thresholds was aggregated at the state-level and compared to reported irrigation water withdrawals from the 2013 FRIS (USDA, 2013) (Table A1). The correlation coefficient $R$ between observed $\overline{IWU}$ and $IWW_{FRIS}$ for SMAP is 0.65 when using $thresh_{\Theta} = 0.04$, but much weaker for AMSR2 (0.35) and AS-CAT (0.15). When increasing $thresh_{\Theta}$ to 0.08, $R$ increases for all soil moisture products (0.75 0.47, 0.23 for SMAP, AMSR2, and ASCAT, respectively). $thresh_{\Theta} = 0.12$ further increases median correlations to 0.79, 0.56, and 0.36 for SMAP, AMSR2,

and ASCAT SM respectively. However, with an increase of the correlation coefficient, the bias and RMSD progressively increase. For this reason, the final threshold $thresh_{\Theta} = 0.12$ is a trade-off of optimal correlation, bias, and RMSD.

*Competing interests.*  None





*Acknowledgements.* Felix Zaussinger, Wouter Dorigo and Alexander Gruber received funding from the European Union Seventh Framework Programme (FP7/2007-2013) under grant agreement number 603608, "Global Earth Observation for integrated water resource assessment": eartH2Observe. In addition, Wouter Dorigo acknowledges the TU Wien Wissenschaftspreis. Luca Brocca, Angelica Tarpanelli and Paolo Fillipuchi received funding from the "WACMOS-Irrigation"project under grant agreement number ESA EXPRO RFP/3-14680/16/I-NB. In

5   addition, the authors acknowledge the TU Wien University Library for financial support through its Open Access Funding Program.

*Data availability.* Data from the ESA CCI Landcover project is available under https://www.esa-landcover-cci.org/. CPC US Unified Precipitation data was provided by the NOAA/OAR/ESRL PSD, Boulder, Colorado, USA, from their Web site at https://www.esrl.noaa.gov/psd/. SMAP soil moisture data was downloaded from NSIDC ( O'Neill, P. E., S. Chan, E. G. Njoku, T. Jackson, and R. Bindlish. 2016. SMAP L3 Radiometer Global Daily 36 km EASE-Grid Soil Moisture, Version 4. Boulder, Colorado USA. NASA National Snow and Ice Data Center

10  Distributed Active Archive Center. doi: https://doi.org/10.5067/OBBHQ5W22HME [23.05.2018]). In addition, we want to thank Robin van der Schalie for providing the AMSR2 LPRMv06 soil moisture data and Sebastian Hahn for providing the modified version of the H111 soil moisture product.



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





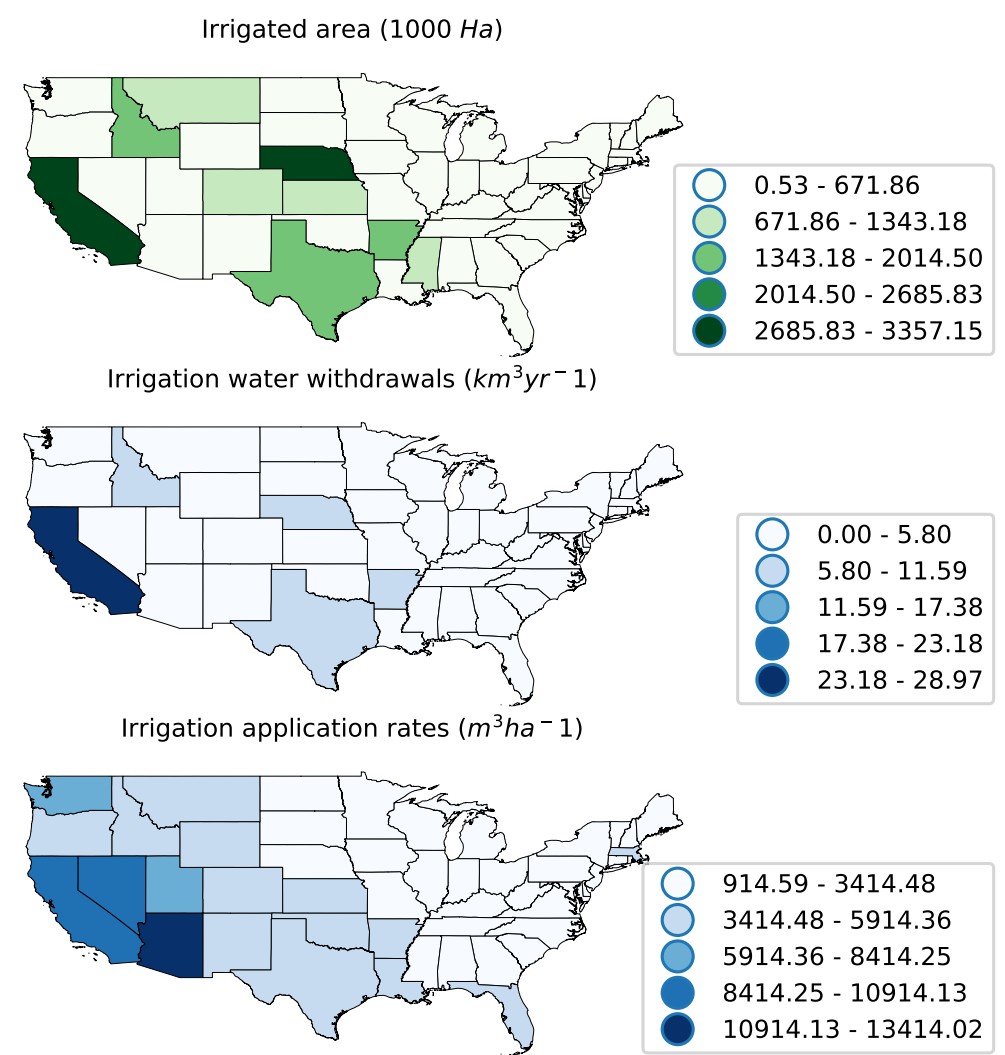

**Figure 1. Per-state irrigated area, irrigation water withdrawals, and irrigation water application rates for 2013.** The data was drawn from the latest Farm and Ranch Irrigation Survey (FRIS) and only reflects irrigation operations in open fields (e.g. excluding crops grown and irrigated in greenhouses).





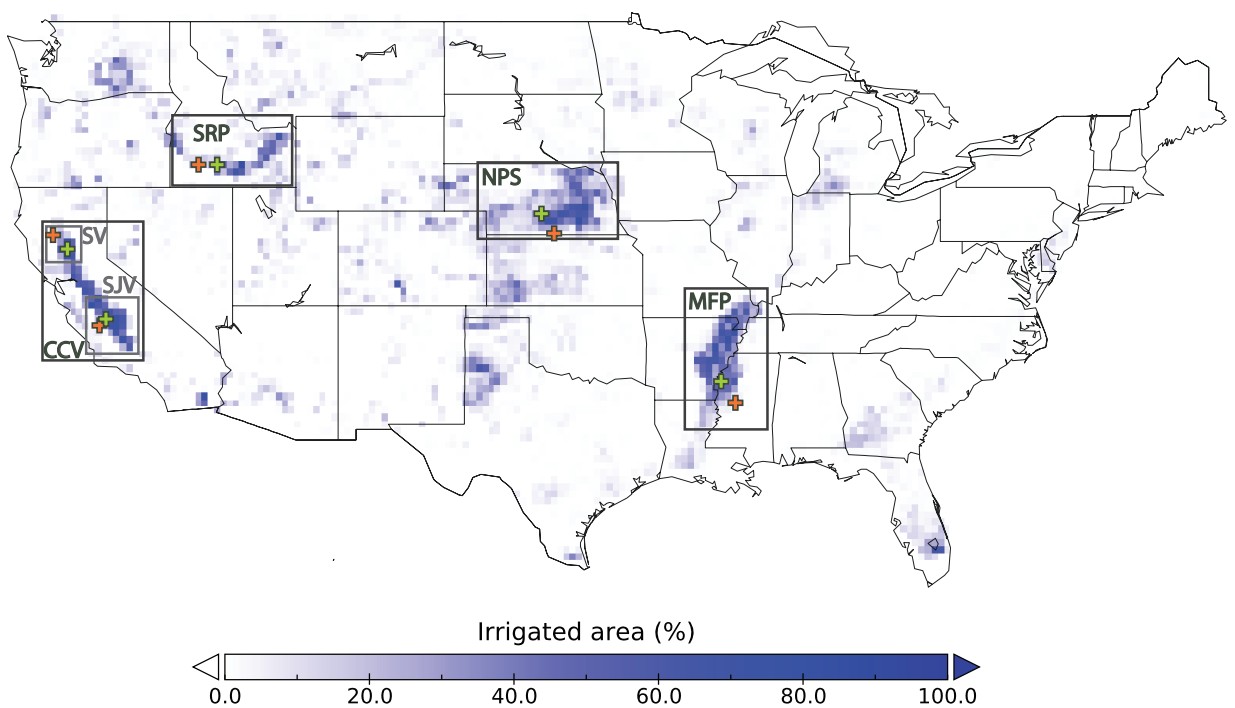

**Figure 2. Study regions and locations of the pixels selected for a local time series analysis over the fractional irrigated area map derived from the spatially aggregated MIrAD-US data set (Pervez and Brown, 2010).** The focus regions are included in black squares and include: the Sacramento Valley (SV) and San Joaquin Valley (SJF) in the California Central Valley (CCV); Snake River Plain (SRP); Nebraska Plains(NP), and the Mississippi Flood Plain (MFP). Green and orange crosses indicate the locations of the irrigated (P-I) and non-irrigated (P-NI) pixels, respectively, at which we further analyze satellite and model soil moisture time series in conjunction with $\overline{IWU}$ estimates.



(a)

(b)

(c)

(d)

(e) Days where $PCP > 0$

(f) Days where $PCP = 0$

**Figure 3. Mean correlation $\bar{r}$ between the daily time series of each satellite soil moisture product (SMAP, AMSR2 and ASCAT) and MERRA-2 soil moisture separated for dry periods (first column; precipitation $PCP > 0$) and dry periods (second column; $PCP = 0$) during the growing season.** Correlations were calculated over agricultural areas only.





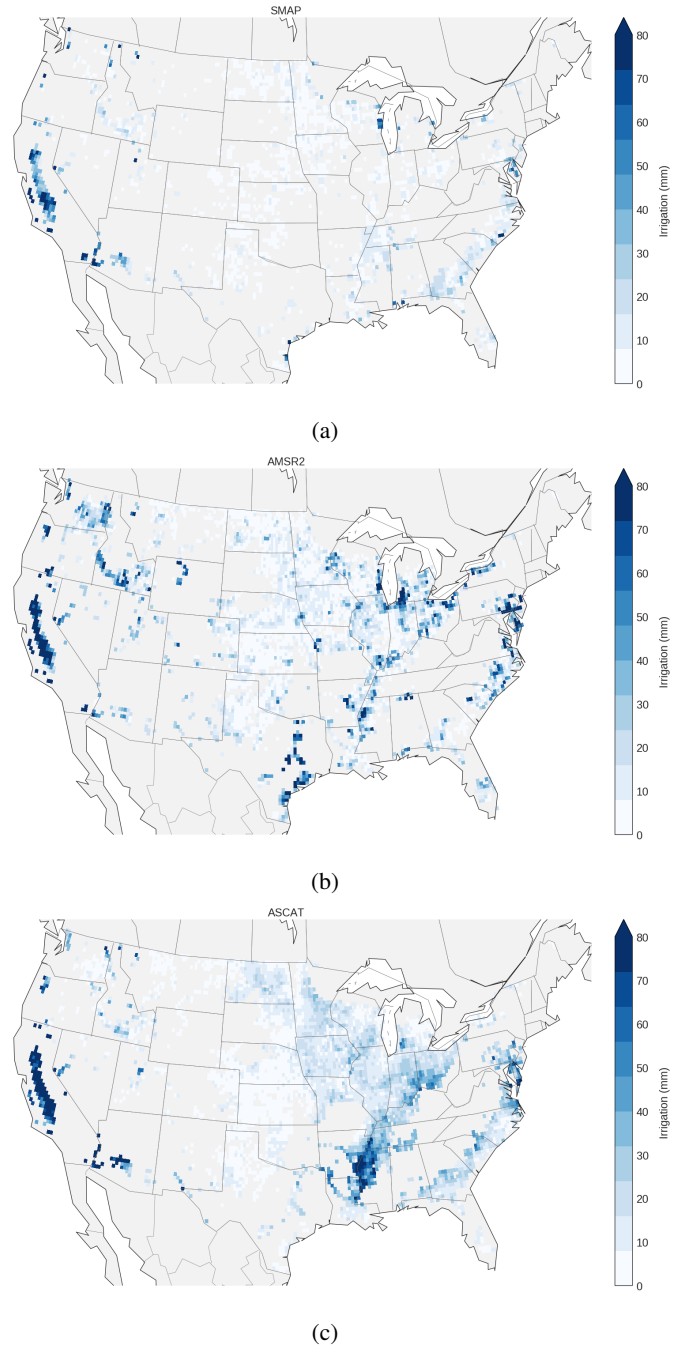

(a)

(b)

(c)

**Figure 4. Mean annual irrigation water use $\overline{IWU}_A$ derived from SMAP (4a), AMSR2 (4b) and ASCAT SM (4c) in combination with modeled SM from MERRA-2.** All pixels with a cropland fraction of $< 5\%$ (as inferred from the CCI land cover product) were excluded from the analysis. Note that for SMAP the climatologies represent the growing season mean of 2015 and 2016, while for AMSR2 and ASCAT the estimates are derived from 4 years of data covering the period of 2013-2016. The upper limit of the colorbar is fixed at the 99th percentile of the SMAP based estimates.


(a) San Joaquin Valley, California

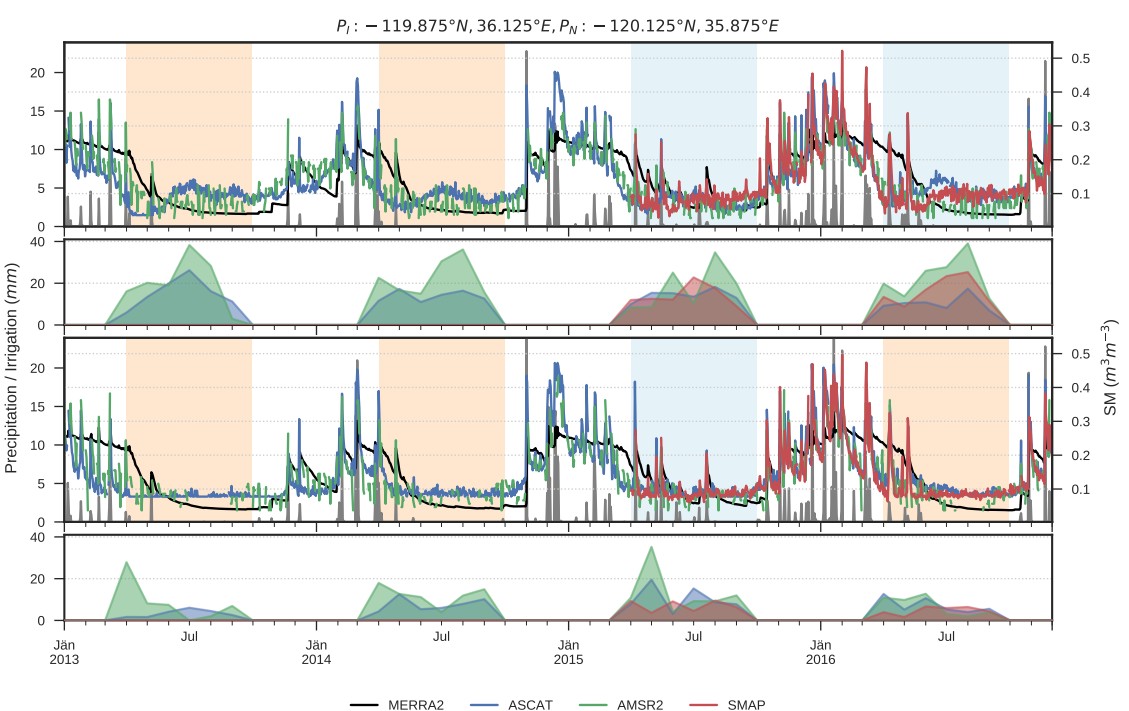

(b) Sacramento Valley, California

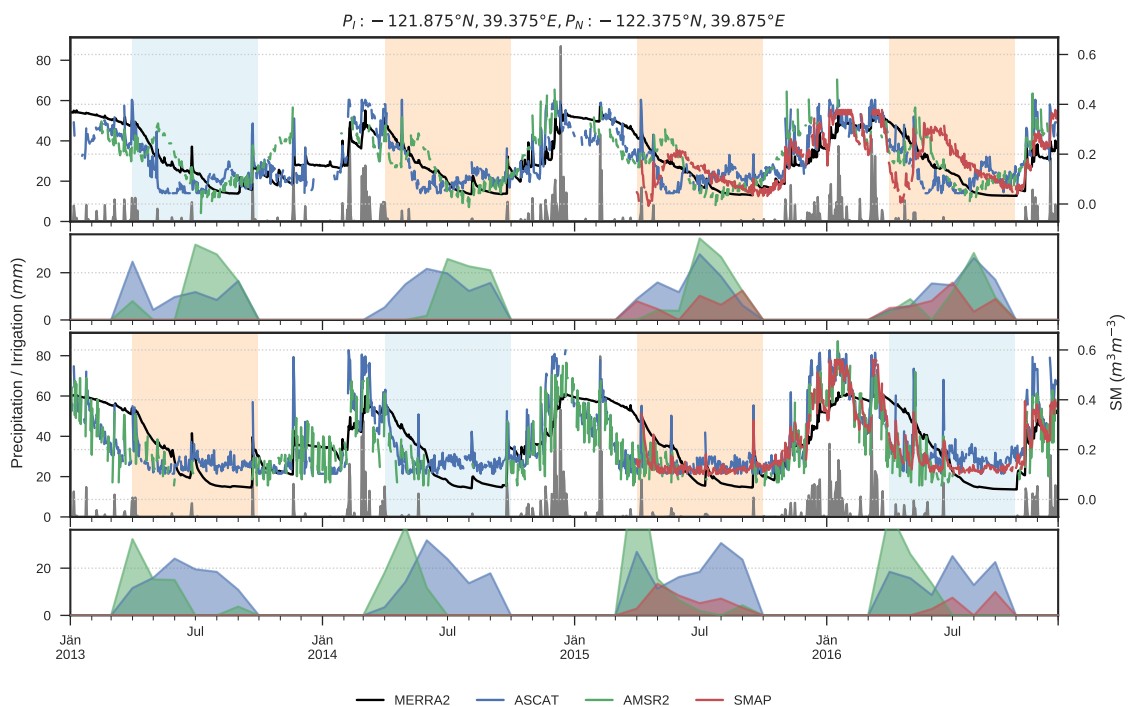



(c) Snake River Plain, Idaho

$P_I: -114.625°N, 42.625°E, P_N: -115.375°N, 42.625°E$

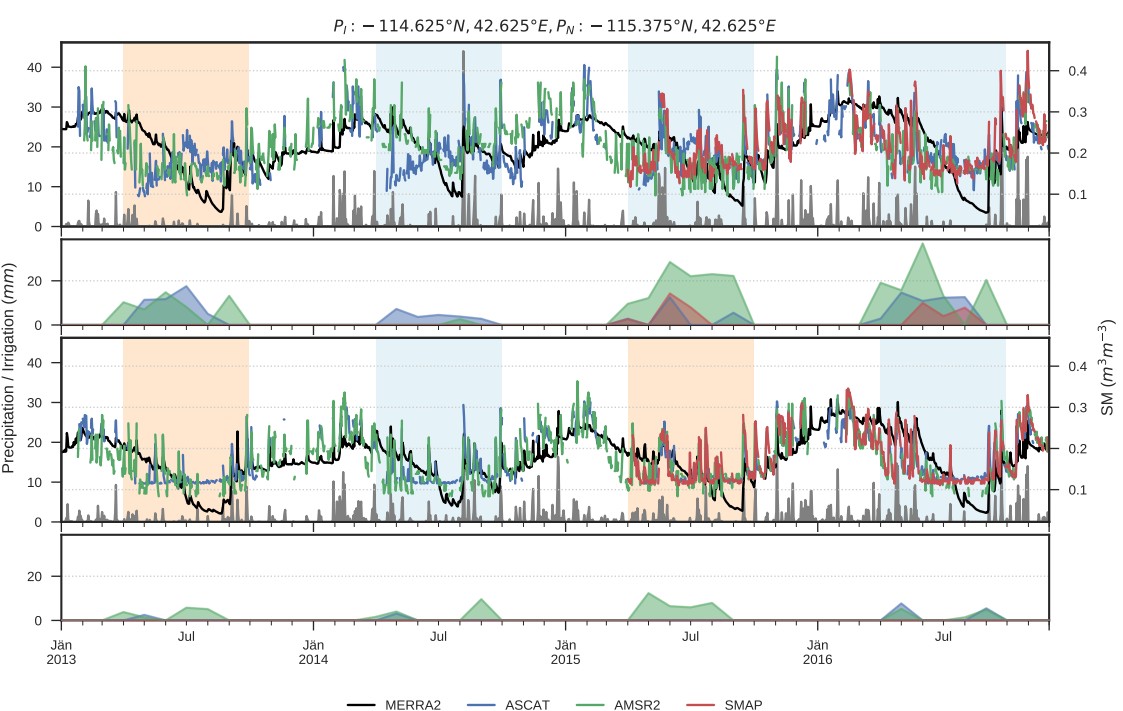

(d) High Plains, Nebraska

$P_I: -99.625°N, 40.625°E, P_N: -99.125°N, 39.875°E$

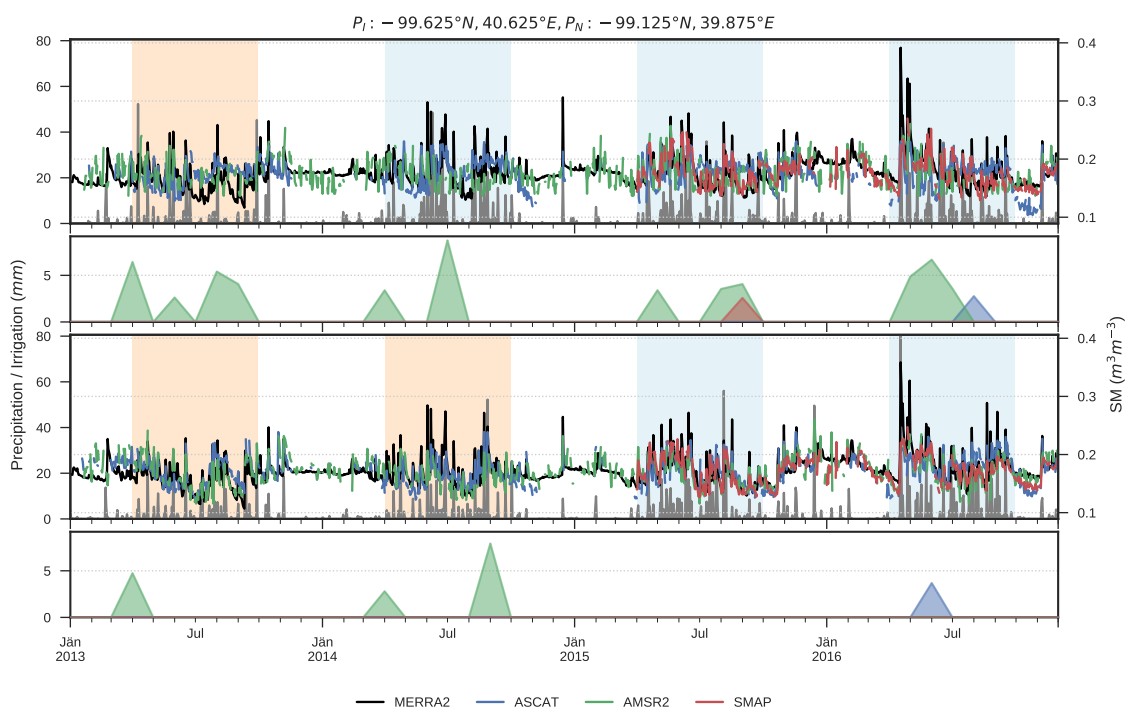



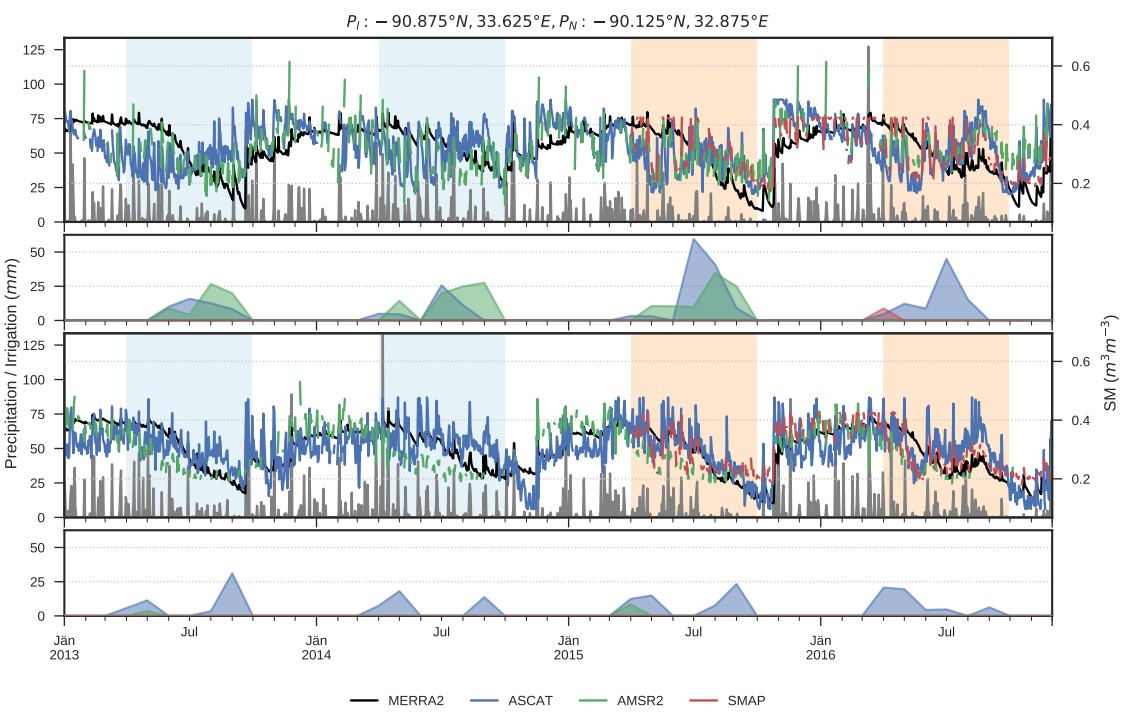

**Figure 5. (continued) Comparison of satellite and model SM time series at irrigated (top two sub-panels) and non-irrigated pixels (bottom two sub-panels) in four regions (figures 5a - 5e).** Daily CPC precipitation is plotted in grey, while blue and orange shadings in the top sub-panels reflect growing seasons with positive and negative rainfall anomalies respectively. The bottom sub-panels show the estimated monthly irrigation water use ($\overline{IWU}_M$) obtained for each satellite-model pair (non-growing season periods have been masked).





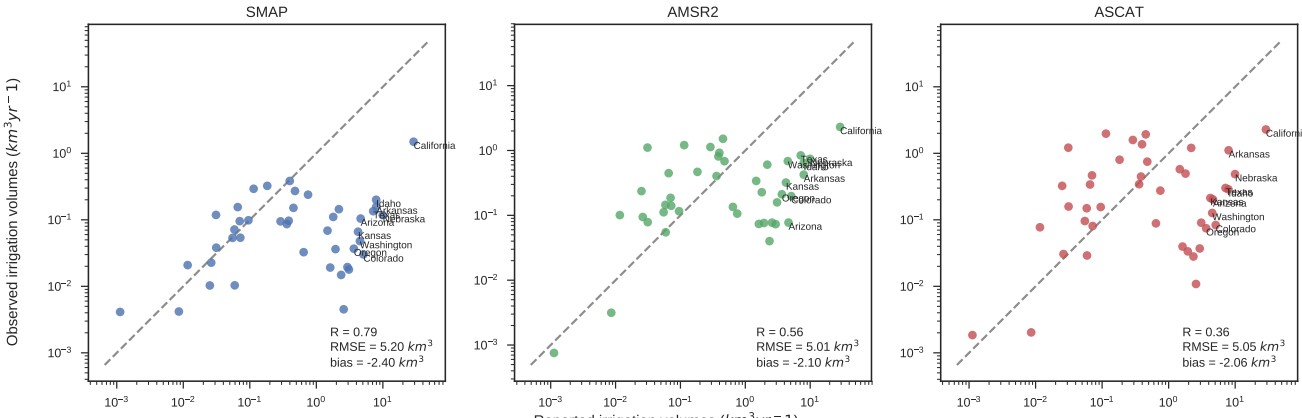

**Figure 6. Comparison of estimated mean annual irrigation water use and reported irrigation water withdrawals.** For each satellite-model pair, observed $\overline{IWU}_A$ was aggregated at the state level. Reported irrigation withdrawals were taken from the 2013 FRIS report and only reflect volumes applied in open fields (e.g. excluding crops grown and irrigated in greenhouses). The data are presented in logarithmic units to reflect both small and large water volumes. Note that the names of the 10 states accounting for the highest irrigation water withdrawals are annotated. $R$, $RMSE$ and $bias$ between observed and reported estimates are shown in the bottom right of each subplot.





**Table 1.** Data products

| Data | Product name | Data availability/ Reference time | Temporal resolution | Spatial resolution | Native gridding | Units |
|---|---|---|---|---|---|---|
| SMAP | SMAP_L3_v4 | 04/2015 - present | 1-3 days | ~40 km | 36 km | $cm^3\,cm^{-3}$ |
| AMSR2 | LPRM v06 | 07/2012 - present | 1-2 days | ~25 km | 0.25° | $m^3\,m^{-3}$ |
| ASCAT | modified H111 | 01/2007 - present | ~1.5 days | ~25 km | 12.5 km | % |
| MERRA-2 | M2T1NXLND.5.12.4 (SFMC parameter) | 01/1980 - present | 1 day (resampled) | ~50 km | 0.625° x 0.5° | $m^3\,m^{-3}$ |
| ESA CCI Land Cover | LC map 2010 epoch | 2008-2012 | - | 0.25° (resampled) | 300 m | - |
| CPC Precipitation | Unified Gauge-Based Analysis of Daily Precipitation | 01/2007 - 12/2016 | 1 day | 0.25° | 0.25° | mm/day |
| MIrAD-US | MODIS Irrigated Agriculture Dataset for the United States v3 | 2012 | - | 0.25° (resampled) | 250 m | - |



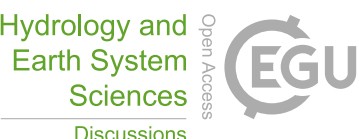

**Table 2. Accuracy assessment of irrigated area estimates.** For each satellite-model combination a confusion matrix between observed $\overline{IWU}_A$ and reference irrigated area from the spatially aggregated 2012 MIrAD-US data set was computed after converting the continuous data to a binary representation (pixels with observed $\overline{IWU}_A \geq 20mm$ and reported $A_{irrigated} \geq 5\%$ were respectively assigned to the irrigated classes). Results are shown for the four states selected in the regional analysis and the contiguous US (CONUS). Underlines indicate the best scores within each region, while bold scores show the overall best.

| Region | Satellite SM product | Error of omission (%) | Error of comission (%) | Overall accuracy (%) | Kappa score (-) |
|---|---|---|---|---|---|
| California | SMAP | 4.94 | 21.43 | **77.68** | **0.33** |
| | AMSR2 | 12.35 | 26.04 | 68.75 | 0.08 |
| | ASCAT | 4.94 | 24.51 | 74.11 | 0.18 |
| Idaho | SMAP | 90.16 | 45.45 | 50.82 | 0.02 |
| | AMSR2 | 40.98 | 40.98 | 59.02 | 0.18 |
| | ASCAT | 77.05 | 39.13 | 54.1 | 0.08 |
| Nebraska | SMAP | 100 | **0** | 19.01 | 0 |
| | AMSR2 | 95.92 | 42.86 | 19.83 | -0.04 |
| | ASCAT | 99.49 | 50 | 19.01 | -0.01 |
| Mississippi | SMAP | 100 | 100 | 60 | -0.11 |
| | AMSR2 | 54.84 | 39.13 | 71.11 | 0.32 |
| | ASCAT | **0** | 62.65 | 42.22 | 0.08 |
| CONUS | SMAP | 90.05 | 53.64 | 74.03 | 0.08 |
| | AMSR2 | 72.7 | 67.64 | 66.82 | 0.08 |
| | ASCAT | 74.75 | 74.65 | 61.88 | 0 |





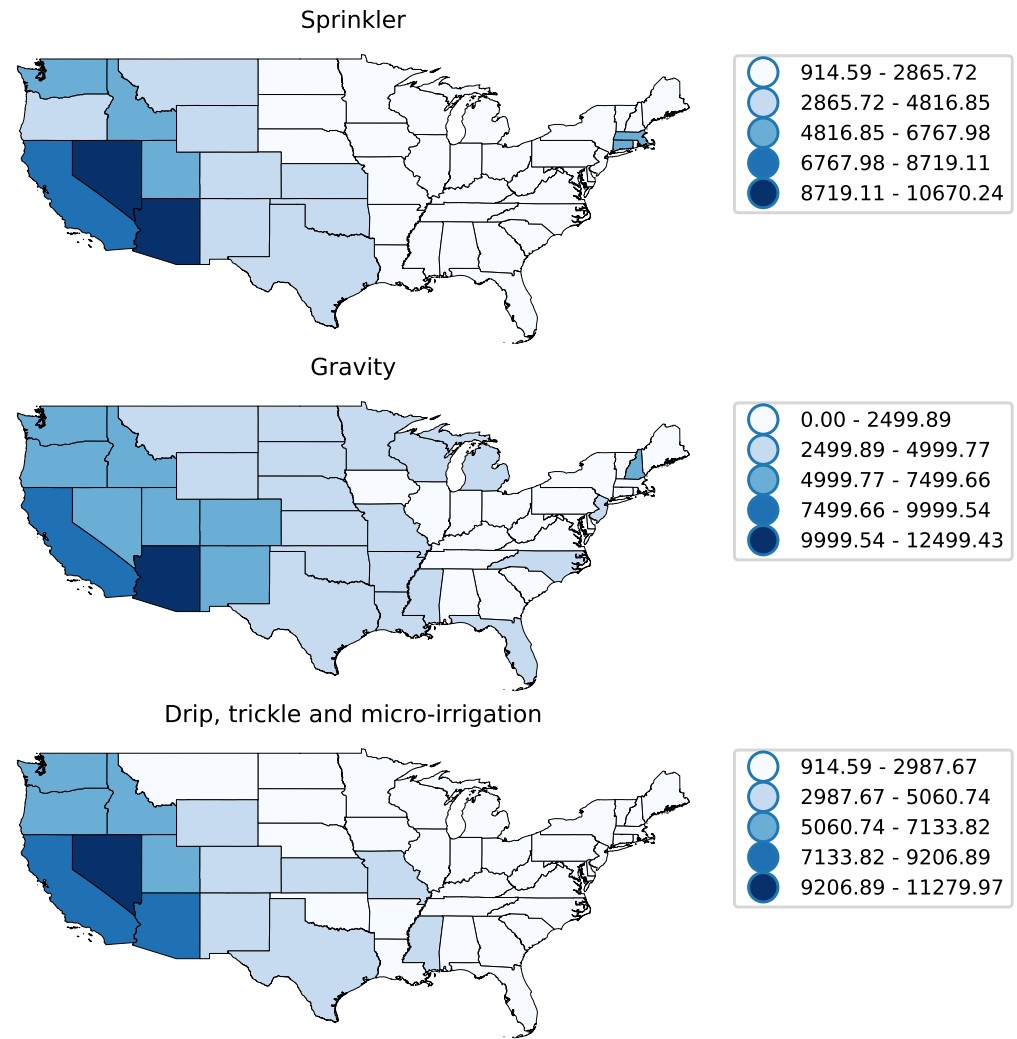

**Figure A1. Per-state irrigation water application rates (m³ha⁻¹) by irrigation technique for 2013.** In accordance with figure 1, the data was derived from the 2013 FRIS and only reflects irrigation operations in open fields.







(a) San Joaquin Valley, California

(b) Snake River Plain, Idaho

(c) Nebraska plains

(d) Mississippi Flood Plain

**Figure A2. Local optimisation of the noise threshold** $thresh_\Theta$**.** For each focus area, the relative differences in irrigation water use ($\overline{IWU}$) estimated at a representative pixel covering a non-irrigated (PNI) and irrigated pixel (PI), respectively ($\frac{\overline{IWU}_{PNI} - \overline{IWU}_{PI}}{\overline{IWU}_{PI}}$) are plotted against $thresh_\Theta$ choices of $0 - 0.2$.





**Table A1. Sensitivity of IWU to optimisation of $tresh_\Theta$ for the entire CONUS.** State-level agreement between estimated annual mean IWU and reference irrigation water withdrawals reported by the 2013 FRIS. The noise threshold $tresh_\Theta$ is applied to the relative increases in satellite soil moisture $\frac{d\Theta_{sat}}{\Theta_{t-n}^{sat}}$ in rain-free periods. Underlined performance scores indicate the best scores within each category.

| Threshold | satellite | R | RMSD $(km^3)$ | bias $(km^3)$ |
|---|---|---|---|---|
| $\frac{d\Theta_{sat}}{\Theta_{t-n}^{sat}} \geqslant 4\%$ | SMAP | 0.65 | 5.09 | -2.26 |
| | AMSR2 | 0.35 | 4.95 | -1.85 |
| | ASCAT | 0.15 | 4.99 | -1.48 |
| $\frac{d\Theta_{sat}}{\Theta_{t-n}^{sat}} \geqslant 8\%$ | SMAP | 0.75 | 5.15 | -2.35 |
| | AMSR2 | 0.47 | 4.98 | -2.00 |
| | ASCAT | 0.23 | 5.01 | -1.82 |
| $\frac{d\Theta_{sat}}{\Theta_{t-n}^{sat}} \geqslant 12\%$ | SMAP | 0.79 | 5.2 | -2.4 |
| | AMSR2 | 0.56 | 5.01 | -2.1 |
| | ASCAT | 0.36 | 5.05 | -2.06 |