# Peer review of "Estimating irrigation water use over the contiguous United States by combining satellite and reanalysis soil moisture data"

_Hydrology and Earth System Sciences, 2018_

## Referee Comment (RC1) · Anonymous Referee #1 · 16 Sep 2018

This article presents an assessment of passive microwave based soil moisture retrievals for irrigation detection. The manuscript is highly relevant and is written well. My comments are listed below.

MAJOR COMMENTS

Section 1.3: The obvious question here is why SMOS is not included in this list. After all, SMOS and SMAP use the L-band instrument, which is supposed to be more sensitive to soil moisture than C and X-band. I think it is essential that SMOS retrievals are included in this comparison for the sake of completeness. If SMOS does a poor job in detecting irrigation, that is also important to quantify and report.

[Figure]

Section 3.4: I am concerned about the use of the normal deviate based rescaling. As shown in Kumar et al. 2015 (HESS), when rescaling is performed relative to the model, it can lead to loss of information. I understand the need to have the datasets in a same space, but that can be done by scaling them using their own mean/standard deviations (see the strategy in Kumar et al. 2015). Using the model's standard deviation for scaling will have a significant impact on the anomalies of the rescaled time series. These analysis should be redone without rescaling to the model's mean/stdev.

The assumption of a reliable model background is very key to this analysis. Generally the NLDAS2 data products are considered to be the "gold standard" over the US where the models are forced with precipitation data informed by gauge+radar information. The choice of MERRA2 is sub-optimal in my opinion. Why not use NLDAS2 datasets that are freely available, instead of MERRA2 (which is also coarser in spatial resolution)?

Section 3.1.1: Since the article was submitted, SMAP released a new version of the data (including L3) that is supposed to have different bias characteristics, in particular. Normally I wouldn't advocate chasing after different versions, but in this case, it is important to use this new version. Since the SMAP data formats haven't changed, I assume this is relatively an easy thing to do.

MINOR COMMENTS:

Page 2, line 5: Correct the quotations – physically "ideal"amount

Page 5, line 16: It'll be good to briefly mention why the global maps differ.

Page 6: With regard to thermal remote sensing, it'll be good to include the Hain et al. JHM 2015 reference ( https://journals.ametsoc.org/doi/full/10.1175/JHM-D-14-0017.1)

Page 7, lines 26-27, fix the quotations.

Figure 2 caption: Please change SJF to SJV, NP to NPS

Page 10, Line 30: MERRA2 used a surface soil moisture layer of 5cm and not 10cm.

Page 10, Lines 31-32: This statement is not strictly true. GLDAS for example, does not assimilate any screen level data.

Figure 3 is nice. Can you show a difference map so it is easy to see where there are significant differences between rdry and rwet? The caption says 'correlations are computed over agricultural areas only'. In that case, why is the map filled everywhere over CONUS?

Page 14, line 27: Is it a known fact that rice vegetation causes specular reflection? Any reference to back up this statement?

Section 5.2 and Figure 4: Can you attribute the rainfall seasonality as another reason why satellite data doesn't seem to detect irrigation. For SMAP, only about 2 years are included. Out of the two, 2016 is considered a wet year over the Midwest, which means there may not have been many days with differences between model and satellite data. Can you provide a figure that indicates the number of days that went into Figure 4, which could help to explain the role of rainfall seasonality.

Figure 5: The choice of the single point sounds a bit arbitrary. Why not, say, do an average of the pixels with irrigated area > 70% (Figure 1). That way you have some spatial representativeness in these time series comparisons.

―――――――――――――

---

## Referee Comment (RC2) · Anonymous Referee #2 · 2 Oct 2018

Review of Zaussinger et al., 'Estimating irrigation water use over the contiguous United States by combining satellite and reanalysis soil moisture data'

With great interest I have read this study trying to quantify irrigation amount from spatial remote sensing and land surface reanalysis. Manuscript is well written in good English and clear, results are well supported by figures, title and abstract reflect the work presented. It is relevant for HESS Readers. I suggest that Authors do some more bibliography as I am missing some relevant work on modelling (and even assimilation) as well as mapping of irrigated areas from the introduction. I also recommend Authors to have a careful read as e.g, many acronyms are missing. It is a very interesting study,

stemming on previous work from e.g. Kumar et al., 2015, but yet I am not completely convinced (and Authors will have to prove me wrong), that is why my recommendation is major review. Please find below an attempt to help.

P.2, L11-13: "It influences the surface wateera5_hres010 r and energy balance through directly increasing soil moisture, which in turn modulates the partitioning of energy between sensible and latent heat (Seneviratne et al., 2010)." I find this sentence slightly simple, is the link with the reference fully appropriate? Are Seneviratne et al., 2010 clearly mentioning irrigation or are they indicating soil moisture in general?, OK soil moisture is increased on irrigated land but water has to come from somewhere else right? Sometimes it is sourced nearby, sometimes not. I think it has to be reflected in the text.

P.2, L.31: "To date, irrigation practices are typically not explicitly included in land surface, [...]" you are right, however some studies have emerged from a modelling point of view (e.g. work from Lawstone et al., 2015 over the US) and even data assimilation point of view (see recent work from Kumar et al., 2018 on the US NLDAS for irrigation intensity over CONUS). I believe it has to be acknowledge in your study.

Lawston PM, Santanello JA Jr, Zaitchik BF, Rodell M (2015) Impact of irrigation methods on land surface model spinup and initialization of WRF forecasts. J Hydrometeorol 16(3):1135–1154

Kumar, S. V., Jasinski, M., Mocko, D., Rodell, M., Borak, J., Li, B., Kato Beaudoing, H., and Peters-Lidard, C. D.: NCA-LDAS land analysis: Development and performance of a multisensor, multi-variate land data assimilation system for the National Climate Assessment, J. Hydrometeor., https://doi.org/10.1175/JHM-D-17-0125.1, online first, 2018.

P.3, section 1.1 on Statistics on irrigated areas and water withdrawals, I am surprised that some more recent work from e.g. Siebert et al. is not mentioned here, please see Siebert et al., 2015 and Meier et al., 2018.

[Figure]

Siebert, S., Kummu, M., Porkka, M., Döll, P., Ramankutty, N., and Scanlon, B. R.: A global data set of the extent of irrigated land from 1900 to 2005, Hydrol. Earth Syst. Sci., 19, 1521-1545, https://doi.org/10.5194/hess-19-1521-2015, 2015.

Meier, J., Zabel, F., and Mauser, W.: A global approach to estimate irrigated areas – a comparison between different data and statistics, Hydrol. Earth Syst. Sci., 22, 1119-1133, https://doi.org/10.5194/hess-22-1119-2018, 2018.

P.4, L.15, please rephrase and extend or remove, it can be better introduced.

P.4, L.25, please provide acronym for USGS (anf later in the text CRU, DEM, ISBA, SURFEX...all acronyms have to be explained).

P.6, L.11-14, this sentence caught my interest, assuming that microwave remote sensing is sensitive to the very first cm of soil, at least at X- and C-band, could you detect all irrigation types? Is dripping/micro-irrigation leading to a sufficient change in soil moisture to be noticed from space? What about vegetation masking the ground? We all have in mind pictures of irrigated (fully developed) corn, can remote sensing see through that? Although it is mentioned in 5.1.3 maybe it could ruled out C-band? At least for certain period of vegetation cycle. By the way, why not considering SMOS L-band mission, my understanding is that it should be more sensitive to soil moisture than the sensors you use (?). SMAP is used but in combination with SMOS (I am aware that combined products exist) it would lead to a longer period being investigated.

P.6, L.27, please add reference for SURFEX (Masson et al., 2013).

Masson, V., Le Moigne, P., Martin, E., Faroux, S., Alias, A., Alkama, R., Belamari, S., Barbu, A., Boone, A., Bouyssel, F., Brousseau, P., Brun, E., Calvet, J.-C., Carrer, D., Decharme, B., Delire, C., Donier, S., Essaouini, K., Gibelin, A.-L., Giordani, H., Habets, F., Jidane, M., Kerdraon, G., Kourzeneva, E., Lafaysse, M., Lafont, S., Lebeaupin Brossier, C., Lemonsu, A., Mahfouf, J.-F., Marguinaud, P., Mokhtari, M., Morin, S., Pigeon, G., Salgado, R., Seity, Y., Taillefer, F., Tanguy, G., Tulet, P., Vincendon, B.,

[Figure]

Vionnet, V., and Voldoire, A.: The SURFEXv7.2 land and ocean surface platform for coupled or offline simulation of earth surface variables and fluxes, Geosci. Model Dev., 6, 929–960, https://doi.org/10.5194/gmd-6-929-2013, 2013.

P.6, L.32, "[...] that SMAP soil moisture carries a clear irrigation signal from rice irrigation [...]", could you please specify irrigation type, flooding (as it is likely to be water seeded)? Did SMAP sees open-water there?

P.11, L.4, didn't you say earlier that this map was not reliable? Please clarify.

Section 3.4, then considering SMOS would make it possible investigating longer period of time.

P.11, L.18-23, please see work from Kumar et al., 2015 on the scaling issue.

P.12, sections 4.1 to 4.3, so a big assumption is that the mismatch between model and satellite soil moisture is irrigation, so the forcing is assumed to be perfect and we know that is not the case. I am also curious on the possible mismatch between what land cover the satellite is really sensing and what MERRA2 has for land cover. The same is true for soil texture, porosity and all ancillary data...could it lead to spurious irrigations/non-irrigation? And what about temporal mismatch? Do you consider the satellite soil moisture revisit time enough for such study? If it rains after the satellite has passed (but maybe it a silly thought as you consider rain free period -according to the forcing)? Please comment on this issue. You should also assess your method in areas where crops are rain-fed only to see what signal is detected when we know that no irrigation occurs (see 5.1.3!)

P.18, L.7, typo (?) "due to"

P.21, L.7-8, "Consequently, microwave soil moisture retrievals are expected to be most sensitive to flood irrigation, followed by sprinkler- and micro-irrigation [...]" are microwaves less sensitive to micro-irrigation or not at all?

Pleas reshape figure 6 as text is hardly readable (and label panels as much as possible

for sack of clarity)

---

## Author Comment (AC1) · 3 Dec 2018

Dear reviewer,

we thank you very much for taking the time and making the effort to review this manuscript. Your feedback was highly constructive and we have tried to incorporate your suggestions to our best knowledge. Attached you can find a zip-file containing a) our responses to your comments, b) a revised version of the manuscript and c) a file showing all the changes made after the initial submission version.

Kind regards, Felix Zaussinger on behalf of the co-authors

[Figure]

Please also note the supplement to this comment:
https://www.hydrol-earth-syst-sci-discuss.net/hess-2018-388/hess-2018-388-AC1-supplement.zip

———————————————————

---

## Author Response (AR1)

**Comments from the reviewers:**

**Reviewer A:**

**"This article presents an assessment of passive microwave based soil moisture retrievals for irrigation detection. The manuscript is highly relevant and is written well."**

**Reply:**

We thank the reviewer for the general endorsement in publishing this manuscript and highly appreciate the detailed and constructive assessment.
* * *
**A.1.** **"Section 1.3: The obvious question here is why SMOS is not included in this list. After all, SMOS and SMAP use the L-band instrument, which is supposed to be more sensitive to soil moisture than C and X-band. I think it is essential that SMOS retrievals are included in this comparison for the sake of completeness. If SMOS does a poor job in detecting irrigation, that is also important to quantify and report."**

**Reply:**

We acknowledge that including soil moisture retrievals from SMOS would be a valuable addition to the manuscript. However, it is not absolutely integral to the completeness of this study. We think that this decision is supported by the following arguments:

1) Our efforts did not aim at conducting a comprehensive assessment of all current microwave soil moisture data sets with respect to irrigation quantification. Actually, our intention is to mainly discuss i) the difference between C-band and L-band, and ii) the difference between active and passive microwave remote sensing. For this reason, we included only three sensors, although with very different properties. For an intercomparison of the performance of various sensors with respect to soil moisture retrieval, we refer to existing literature (see point 2).
2) Soil moisture retrievals based on SMAP have repeatedly been shown to be more accurate than those based on SMOS [1,2].
3) The performance of SMOS soil moisture retrievals for irrigation quantification is thoroughly discussed in very recently published work [3]. In this paper, it was found that SMAP, ASCAT and AMSR-2 soil moisture outperform SMOS soil moisture with respect to rainfall estimation via SM2RAIN and that SMAP performs best overall. We argue that this is a viable indicator for the corresponding capabilities of the products in quantifying irrigation (since it can be simply added to precipitation), providing evidence that it is sufficient to solely include a single high quality L-band product.
4) Regarding the sensitivity of C- and L-band microwave observations to soil moisture, it has been shown that under certain conditions (e.g., dense vegetation cover), C-band soil moisture retrievals can be equally or even more accurate as L-band retrievals [4,5].

**Changes in manuscript:**

We added the following sentence to anticipate the obvious question why SMOS is not included in the analysis:

"By using passive L-band and both active and passive C-band soil moisture data, we aim to assess the impact of the microwave observation frequency and the sensing technique with respect to irrigation quantification. For this reason we only used one dataset per category."

**References:**
[1] Colliander, A., Jackson, T. J., Bindlish, R., Chan, S., Das, N., Kim, S. B., ... & Asanuma, J. (2017). Validation of SMAP surface soil moisture products with core validation sites. Remote sensing of environment, 191, 215-231.
[2] Chan, S. K., Bindlish, R., O'Neill, P. E., Njoku, E., Jackson, T., Colliander, A., ... & Yueh, S. (2016). Assessment of the SMAP passive soil moisture product. IEEE Transactions on Geoscience and Remote Sensing, 54(8), 4994-5007.
[3] Brocca, L., Tarpanelli, A., Filippucci, P., Dorigo, W., Zaussinger, F., Gruber, A., & Fernández-Prieto, D. (2018). How much water is used for irrigation? A new approach exploiting coarse resolution satellite soil moisture products. International Journal of Applied Earth Observation and Geoinformation, 73, 752-766.
[4] Al-Yaari, A., Wigneron, J. P., Ducharne, A., Kerr, Y. H., Wagner, W., De Lannoy, G., ... & Mialon, A. (2014). Global-scale comparison of passive (SMOS) and active (ASCAT) satellite based microwave soil moisture retrievals with soil moisture simulations (MERRA-Land). Remote Sensing of Environment, 152, 614-626.
[5] van der Schalie, R., de Jeu, R., Parinussa, R., Rodríguez-Fernández, N., Kerr, Y., Al-Yaari, A., ... & Drusch, M. (2018). The Effect of Three Different Data Fusion Approaches on the Quality of Soil Moisture Retrievals from Multiple Passive Microwave Sensors. Remote Sensing, 10(1), 107.
* * *
**A.2.** "Section 3.4: I am concerned about the use of the normal deviate based rescaling. As shown in Kumar et al. 2015 (HESS), when rescaling is performed relative to the model, it can lead to loss of information. I understand the need to have the datasets in a same space, but that can be done by scaling them using their own mean/standard deviations (see the strategy in Kumar et al. 2015). Using the model's standard deviation for scaling will have a significant impact on the anomalies of the rescaled time series. These analysis should be redone without rescaling to the model's mean/stdev."

**Reply:**
This is a very valuable criticism. However, apart from having the data in the same space, we do need the rescaling to correct for differing representative layer depths and spatial resolutions between products. It is true that the rescaling does have a significant impact on the anomalies, but we purposefully want that impact. Without the rescaling to the models mean and standard deviation, we could not justify a direct comparison of the irrigation estimates based on the respective soil moisture products.

Specifically, by matching the temporal mean and standard deviation of the satellite data sets to the model data set we correct for both horizontal and vertical systematic representativeness errors. This technique has been employed by various studies, e.g., [1,2]. In addition, we hereby implicitly compensate for different units (i.e., volumetric soil moisture

and degrees of saturation), which also avoids explicit conversion using (often inaccurate) auxiliary data (e.g., soil porosity maps).

**Changes in manuscript:**
None

**References:**
[1] Dorigo, W., Scipal, K., Parinussa, R. M., Liu, Y. Y., Wagner, W., De Jeu, R. A., & Naeimi, V. (2010). Error characterisation of global active and passive microwave soil moisture data sets.
[2] Albergel, C., De Rosnay, P., Gruhier, C., Muñoz-Sabater, J., Hasenauer, S., Isaksen, L., ... & Wagner, W. (2012). Evaluation of remotely sensed and modelled soil moisture products using global ground-based in situ observations. Remote Sensing of Environment, 118, 215-226.
* * *
**A.3.** **"The assumption of a reliable model background is very key to this analysis. Generally the NLDAS2 data products are considered to be the "gold standard" over the US where the models are forced with precipitation data informed by gauge+radar information. The choice of MERRA2 is sub-optimal in my opinion. Why not use NLDAS2 datasets that are freely available, instead of MERRA2 (which is also coarser in spatial resolution)?"**

**Reply:**
It is true that the quality of the model soil moisture estimates is integral to the proposed methodology. However, most products, including NLDAS2, integrate observations of surface temperature and surface humidity. As outlined in [1] and [2], these observations are indirectly impacted by irrigation practices and hence implicitly alter soil moisture simulations. Although NLDAS2 would be a very suitable model choice in terms of quality, integrity and spatio-temporal resolution, we cannot employ it because near-surface air temperature and specific humidity are included in the forcing data [3].

**Changes in manuscript:**
None

**References:**
[1] Wei, J., Dirmeyer, P. A., Wisser, D., Bosilovich, M. G., and Mocko, D. M.: Where Does the Irrigation Water Go? An Estimate of the Contribution of Irrigation to Precipitation Using MERRA, Journal of Hydrometeorology, 14, 275–289, https://doi.org/10.1175/jhm-d-12-079.1, http://dx.doi.org/10.1175/JHM-D-12-079.1, 2013.
[2] Tuinenburg, O. and Vries, J.: Irrigation Patterns Resemble ERA-Interim Reanalysis Soil Moisture Additions, Geophysical Research Letters, 44, 2017.
[3] https://ldas.gsfc.nasa.gov/nldas/NLDAS2forcing.php
* * *
**A.4.** **"Section 3.1.1: Since the article was submitted, SMAP released a new version of the data (including L3) that is supposed to have different bias characteristics, in particular. Normally I wouldn't advocate chasing after different versions, but in this case, it is important to use this new version. Since the SMAP data formats haven't changed, I assume this is relatively an easy thing to do."**

**Reply:**
We thank you very much for pointing out the new version release. We collectively agree with you that it is crucial to use the updated SMAP Passive L3 V5 product. We investigated the impact of switching to V5 and found that the newer version indeed addresses the dry bias observed in V4. Moreover, the ascending and descending observations are more homogenous amongst each other. In the end, we re-processed all results using the SMAP V5 soil moisture data set.

**Changes in manuscript:**
First, we changed part of the data section to describe the key characteristics of V5.

Section 3.1.1, Page 11, Line 4
We used the L3_SM_P V5 data product, which is sampled at 36 km resolution. In this product version, a water body correction and an improved soil temperature depth correction have been applied, which have respectively reduced anomalous soil moisture values near large water bodies and the dry bias with respect to the SMAP core validation sites (Jackson, 2018). [1]

Moreover, all results (i.e., tables and plots) and numbers in the text were updated accordingly.

**References:**
[1]https://nsidc.org/sites/nsidc.org/files/technical-references/L2SMPE_Asmt_Rpt_EOPM_v5c_Jun2018.pdf
* * *
**A.5.** **"Page 2, line 5: Correct the quotations – physically "ideal"amount"**

**Reply:**
Thank you for pointing that out, we have corrected the position of the leading quotation mark.

**Changes in manuscript:**
Section 1, Page 3, Line 5: "...physically "ideal" amount."
* * *
**A.6.** **"Page 5, line 16: It'll be good to briefly mention why the global maps differ."**

**Reply:**
We agree that a brief discussion of the reasons for the substantial systematic deviations between different global irrigated area products is worthwhile.

**Changes in manuscript:**
Section 1.2.1, Page 6, Line 16
"However, there are large discrepancies between the different global data sets mainly stemming from varying definitions of irrigated areas among the data sets (i.e. area equipped for irrigation, irrigated area and cropped area), differences in the quality and spatial resolution of the input data sets and differing reference years (i.e., the years 2000 and 2005) (Salmon et al., 2015, Meier et al., 2018)."

**References:** [1] Meier, J., Zabel, F., & Mauser, W. (2018). A global approach to estimate irrigated areas–a comparison between different data and statistics. Hydrology and Earth System Sciences, 22(2), 1119-1133.
* * *
**A.7.** **"Page 6: With regard to thermal remote sensing, it'll be good to include the Hain et al. JHM 2015 reference (https://journals.ametsoc.org/doi/full/10.1175/JHM-D-14-0017.1)"**

**Reply:**
We were not aware of this interesting work and regard it as a valuable addition to the paragraph focusing on thermal remote sensing approaches.

**Changes in manuscript:**
Section 1.2.1, Page 7, Line 9
"In contrast, Hain et al. [1] developed a novel method for inferring regions where non-precipitation inputs (e.g., irrigation) significantly impact terrestrial latent heat flux (LE). They compared modelled bottom-up LE (i.e., without irrigation) and top-down LE drawn from observations of diurnal land surface temperature changes which are connected to changes in the land surface moisture status and therefore irrigation."

**References:**
[1] Hain, C. R., Crow, W. T., Anderson, M. C., & Yilmaz, M. T. (2015). Diagnosing neglected soil moisture source–sink processes via a thermal infrared–based two-source energy balance model. *Journal of Hydrometeorology*, *16*(3), 1070-1086.
* * *
**A.8.** **"Page 7, lines 26-27, fix the quotations."**

**Reply:**
We were not completely sure what was meant by this comment. We interpreted it as referring to a wrong position of the quotation and corrected it correspondingly.

**Changes in manuscript:**
Section 2.1, Page 7, Line 26-28
"The 2013 Farm and Ranch Irrigation Survey (FRIS) of the National Agricultural Statistics Service (NASS) of the USDA provides selected irrigation data from surveys conducted at approximately 35000 farms using irrigation across the US (USDA, 2013)."
* * *
**A.9.** **"Figure 2 caption: Please change SJF to SJV, NP to NPS"**

**Reply:**

Thank you very much for highlighting these errors. The caption of figure 2 now correctly attributes the abbreviations introduced for the focus regions.

**Changes in manuscript:**

Page 31, Figure 2: "...the Sacramento Valley (SV) and San Joaquin Valley (SJV) in the California Central Valley (CCV); Snake River Plain (SRP); Nebraska Plains (NPS), and the Mississippi Flood Plain (MFP)."
* * *
**A.10.** "Page 10, Line 30: MERRA2 used a surface soil moisture layer of 5cm and not 10cm."

**Reply:**

You are perfectly right. Unfortunately, due to a misleading rounding error in the Panoply viewer we assumed a 0.1 m surface soil moisture layer depth. Per default the softwareonly seems to display 1 digit float values, so 0.05m was rounded up to 0.1m [1,2]. We were not aware of this default setting. Apart from updating Page 10, Line 30, this also impacts all results concerning irrigation quantification. Specifically, in the current rescaling formulation, this will directly reduce all IWU estimates by half and will be addressed in the revised manuscript version.

**Changes in manuscript:**

We have corrected Page 12, Line 8 to the actual layer depth of 5 cm. In addition, all IWU estimates where thus reduced by half and were updated accordingly.

**References:**

[1] MERRA-2: File Specification (https://gmao.gsfc.nasa.gov/pubs/docs/Bosilovich785.pdf)
[2] Reichle, R. H., Draper, C. S., Liu, Q., Girotto, M., Mahanama, S. P., Koster, R. D., & De Lannoy, G. J. (2017). Assessment of MERRA-2 land surface hydrology estimates. *Journal of Climate*, *30*(8), 2937-2960. (https://journals.ametsoc.org/doi/10.1175/JCLI-D-16-0720.1)
* * *
**A.11.** "Page 10, Lines 31-32: This statement is not strictly true. GLDAS for example, does not assimilate any screen level data."

**Reply:**

Based on your comment we investigated the GLDAS (Version 2) assimilation scheme. According to the official website [1], a suite of atmospheric data sets are used to force the model. In version 2, the main inputs include the Princeton University meteorological forcing dataset [2], various precipitation data sets based on satellite observations and a range of land surface data sets. Specifically in [2], a summary of all the forcings used to construct the long term data set is given in Table 1. It can be seen that two of the five datasets include observations which can be altered by irrigation. The NCEP–NCAR reanalysis includes surface air temperature (T) and specific humidity (q) observations whereas CRU TS2.0 only includes T (assimilated from station observations). Hence, based on the here presented information, we cannot use GLDAS (V2) as an unbiased reference for the purpose of

irrigation quantification. Needless to say, we are open for other independent references concerning the GLDAS assimilation scheme.

**Changes in manuscript:**
We deleted the Examples "ERA-Interim" and "GLDAS" from the brackets on page 12, line 10.

**References:**
[1] https://ldas.gsfc.nasa.gov/gldas/GLDASforcing.php
[2] Sheffield, J., Goteti, G., & Wood, E. F. (2006). Development of a 50-year high-resolution global dataset of meteorological forcings for land surface modeling. *Journal of Climate*, *19*(13), 3088-3111.
* * *
**A.12.** **"Figure 3 is nice. Can you show a difference map so it is easy to see where there are significant differences between rdry and rwet? The caption says 'correlations are computed over agricultural areas only'. In that case, why is the map filled everywhere over CONUS?"**

**Reply:**
We thank you for stating that figure 3 proves valuable to the manuscript. As requested, we have added a difference map to the Appendix. Regarding the caption, thank you for pointing that out! The caption correctly states that the correlations are only computed over agricultural areas, however, the masking was not performed. Therefore, all other land regions were masked accordingly.

**Changes in manuscript:**
Page 32, Figure 3: The figures were correctly masked for agricultural land cover. Difference maps were added to the Appendix section (Figure A2).
* * *
**A.13.** **"Page 14, line 27: Is it a known fact that rice vegetation causes specular reflection? Any reference to back up this statement?"**

**Reply:**
Here, we carefully need to differentiate between different phenological development phases of rice. Between the actual establishment of the flood irrigation (the fields are usually covered with 10-15 cm of water) and the time when the rice starts to break through the water surface, we assume that specular reflection can be an issue (given low water surface roughness, i.e., low surface wind speeds). In the second period of the rice breaking through the water surface until the fields are ultimately drained for harvest, rice vegetation above the water surface may act as double-bounce/cornerstone reflectors. The characteristic interaction between active C-band microwave signals and rice phenology is discussed in e.g. [1-3] and is considered as well established.

**Changes in manuscript:**
We completely reformulated the argument being made.

"During the early phenological growth phase of rice, this observation can be attributed to specular reflection of the radar signal from the flood water surface, given that wind speeds do not significantly affect the water's surface roughness [2]. By the time the rice stems start to break through the water surface the now elongated rice stems are known to act as double-bounce reflectors, which commonly results in an enhanced backscatter signal that can be observed until field drainage in late summer [1,3] (see ASCAT soil moisture time series in figure 5b)."

**References:**

[1] Le Toan, T., Ribbes, F., Wang, L. F., Floury, N., Ding, K. H., Kong, J. A., ... & Kurosu, T. (1997). Rice crop mapping and monitoring using ERS-1 data based on experiment and modeling results. *IEEE Transactions on Geoscience and Remote Sensing*, *35*(1), 41-56.
[2] Nguyen, D. B., Clauss, K., Cao, S., Naeimi, V., Kuenzer, C., & Wagner, W. (2015). Mapping rice seasonality in the Mekong Delta with multi-year Envisat ASAR WSM data. *Remote Sensing*, 7(12), 15868-15893.
[3] Nguyen, D. B., Gruber, A., & Wagner, W. (2016). Mapping rice extent and cropping scheme in the Mekong Delta using Sentinel-1A data. *Remote Sensing Letters*, 7(12), 1209-1218.
* * *
**A.14.** "**Section 5.2 and Figure 4: Can you attribute the rainfall seasonality as another reason why satellite data doesn't seem to detect irrigation. For SMAP, only about 2 years are included. Out of the two, 2016 is considered a wet year over the Midwest, which means there may not have been many days with differences between model and satellite data. Can you provide a figure that indicates the number of days that went into Figure 4, which could help to explain the role of rainfall seasonality.**"

**Reply:**

Here, the reviewer raises a very important point. Indeed, during 'wet' years we expect that the method is less skillful in detecting and quantifying irrigation events. Since figure 4 indicates the average amount of irrigation per year and for SMAP only 2 years of data are available, we created the requested plot in a normalized fashion: i.e., plotted the number of days with irrigation > 0 (according to the method) normalized by the number of years which went into the overall estimation (4 for ASCAT and AMSR-2, 2 for SMAP). The plot therefore indicates the average number of irrigation events (detected by the method) per year.

**Changes in manuscript:**

We added the new plot(s) to the supplement (Figure A3) and included a brief discussion regarding the role of rainfall seasonality in Section 5.2, Page 18, Lines 23-30:

"Rainfall seasonality is another potential reason for the underestimation in the central U.S, where the climate transitions from arid in the west to humid in the east. To investigate its impact, we plotted the average number of days per growing season where IWU > 0 (figure A3), which sums up to the number of days that went into the IWU estimates shown in figure 4. It can be seen that for SMAP based IWU , a significant number of days with irrigation

(mean count) only is detected in the arid west and south-west. For AMSR2, the mean counts are highest in California, although counts in the range of 20-30 occur in the Snake River Valley, Mississippi Flood Plain and other agricultural regions. In agreement with the passive products, mean counts for ASCAT are highest in California and south-western states. There also is a clear pattern in the Mississippi Valley and along the south-eastern states."
* * *
**A.15.** "Figure 5: The choice of the single point sounds a bit arbitrary. Why not, say, do an average of the pixels with irrigated area > 70% (Figure 1). That way you have some spatial representativeness in these time series comparisons."

**Reply:**
We understand your criticism. However, the fractional irrigated area in itself is not most relevant, as we are interested in the amount of water applied for irrigation. Specifically, we care about entangling conditions at the scale of field districts to investigate the impact of varying regional irrigation practices and crop types as well as climate conditions. Therefore, there would be no added spatial representativeness in averaging all pixels with say irrigated area > 70%, because the individual cases are far too heterogeneous.

**Changes in manuscript:**
None

**Reviewer B:**

**"With great interest I have read this study trying to quantify irrigation amount from spatial remote sensing and land surface reanalysis. Manuscript is well written in good English and clear, results are well supported by figures, title and abstract reflect the work presented. It is relevant for HESS Readers."**

**Reply:**
We thank the reviewer for his positive response. Moreover, we are delighted to hear that for you the manuscript was an interesting read and that you consider it as relevant for the HESS readership.

**"I suggest that Authors do some more bibliography as I am missing some relevant work on modelling (and even assimilation) as well as mapping of irrigated areas from the introduction. I also recommend Authors to have a careful read as e.g, many acronyms are missing. It is a very interesting study, stemming on previous work from e.g. Kumar et al., 2015, but yet I am not completely convinced (and Authors will have to prove me wrong), that is why my recommendation is major review."**

**Reply:**
We thank the reviewer for the overall interest in our study. We have tried to include more literature (including the important suggestions you made) to enhance the discussion of modelling and data assimilation in the context of irrigation quantification. We hope that by now, we have made the necessary steps to convince you.
* * *
**B.1.** **"P.2, L11-13: "It influences the surface water and energy balance through directly increasing soil moisture, which in turn modulates the partitioning of energy between sensible and latent heat (Seneviratne et al., 2010)." I find this sentence slightly simple, is the link with the reference fully appropriate? Are Seneviratne et al., 2010 clearly mentioning irrigation or are they indicating soil moisture in general?, OK soil moisture is increased on irrigated land but water has to come from somewhere else right? Sometimes it is sourced nearby, sometimes not. I think it has to be reflected in the text."**

**Reply:**
We thank you for your criticism regarding the reference of Seneviratne et al. 2010. Indeed, the citation in the first sentence is a bit confusing. We meant to only reference the second part of the sentence (i.e., "..., which in turn modulates the partitioning of energy between sensible and latent heat (Seneviratne et al., 2010)."). However, it reads as if the reference corresponds to the whole sentence, whereas the paper only explicitly refers to soil moisture, as already outlined by yourself. In order to eliminate potential confusion, we chose to divide the sentence into two parts.

**Changes in manuscript:**
Section 1, Page 2, Line 11

"Irrigation influences the surface water and energy balance through directly increasing soil moisture. In turn, soil moisture is widely known to modulate the partitioning of energy between sensible and latent heat (Seneviratne et al., 2010)."
* * *
**B.2.** "P.2, L.31: "To date, irrigation practices are typically not explicitly included in land surface, [...]" you are right, however some studies have emerged from a modelling point of view (e.g. work from Lawstone et al., 2015 over the US) and even data assimilation point of view (see recent work from Kumar et al., 2018 on the US NLDAS for irrigation intensity over CONUS). I believe it has to be acknowledge in your study."

**Reply:**
We thank you for referring to these interesting papers. Indeed, we were not aware enough of recent literature regarding the modelling and assimilation of irrigation data. To address this shortcoming, we have revised and extended the respective paragraph on irrigation modelling and data assimilation by including a discussion of both [1] and [2].

**Changes in manuscript:**
Section 1, Page 2, Line 31 to Page 3, Line 24
"To date, irrigation practices are typically not explicitly included in land surface, climate, or weather models. On the other hand, irrigation directly impacts land surface temperature, humidity, and soil moisture observations, and through them indirectly impact model simulations when they are being assimilated (Tuinenburg and Vries, 2017). A range of climate modelling studies employed irrigation modules on a global scale. Mainly based on a combination of static spatial maps of irrigated area and soil moisture and/or vegetation data, they tried to approximate seasonal IWU (Lobell et al., 2006; Bonfils and Lobell, 2007; Kueppers et al., 2007). However, the simulated impact of irrigation on both global and regional climate showed considerable variation across studies. With respect to a contiguous U.S. domain, Lawston et al. (2015) assessed the effects of drip, flood, and sprinkler irrigation methods during a climatically dry and wet year on land–atmosphere interactions. They used the National Aeronautics and Space Administration's (NASA) high-resolution Land Information System (LIS) and the NASA Unified Weather Research and Forecasting Model (NU-WRF) framework both in offline and coupled simulations. In accordance with previous studies, they found that irrigation indeed cools and moistens the surface over and downwind of irrigated areas. Moreover, they found that the magnitude of this irrigation cooling effect (ICE) strongly depends on the parametrization of the respective irrigation methods. In a very recent study, KUMAR et al. (2018) conducted a multisensor, multivariate land data assimilation experiment over the CONUS by using the NASA LIS to enable the National Climate Assessment (NCA) Land Data Assimilation System (NCA-LDAS). Particularly, the use of a larger-than-normal range of soil moisture data records and snow depth data from microwave remote sensing combined with an irrigation intensity map systematically improved soil moisture and snow depth simulations.

With respect to the discrepancies in the global modelling studies, Sacks et al. (2009) argued that they can be primarily explained by systematic differences in the control of irrigation water application within the respective modules, e.g. by climate, food demand, and

economical conditions. Logically, this arguments also holds true for the modelling study by Lawston et al. (2015). Regarding the irrigation forcing used in KUMAR et al. (2018), we argue that the term "irrigation intensity" gives a false impression. Irrigation intensity in a physical sense should not be attributed to fractional irrigated area, but must rather be connected to the actual irrigation water use per unit area. In addition, fields may be either over- or under-irrigated with respect to the physically "ideal" amount. Hence, current irrigation modules are unable to consistently reflect real-world conditions and thus introduce uncertainties in modelling and data assimilation. Consequently, information on the spatio-temporal distribution and development of actual IWU is needed to improve the representation of land-atmosphere feedbacks in model simulations (Ozdogan et al., 2010a)."

* * *
**B.4.** **"P.4, L.15, please rephrase and extend or remove, it can be better introduced."**

**Reply:**
We agree that at this specific location the introduction of remote sensing for irrigation mapping was too fuzzy. We therefore chose to move (and modify) lines 15-16 to section 1.1, which in our view establishes the connection with section 1.2 without appearing as too short.

**Changes in manuscript:**
We removed the sentence and added a new one to Section 1.1, Page 4, Line 14 in order to link to Section 1.2:
"On the basis of these drawbacks, remote sensing evolved as an effective tool to potentially overcome these limitations since it provides synoptic, independent and timely information of biogeophysical variables that are either directly or indirectly related to irrigation."
* * *
**B.5.** **"P.4, L.25, please provide acronym for USGS (anf later in the text CRU, DEM, ISBA, SURFEX...all acronyms have to be explained)."**

**Reply:**
We thank the reviewer for pointing out that several acronyms were left unexplained. In the revised manuscript version, all acronyms are now correctly introduced.

**Changes in manuscript:**
We have now introduced the missing explanations for each of the following acronyms:
-   USGS : United States Geological Survey
-   CRU : Climate Research Unit
-   DEM : Digital Elevation Model
-   ISBA : Interaction Sol Biosphère Atmosphère
-   SURFEX : Surface Externalisée

**References:**
Explanations for each acronym were provided within the respective literature.
* * *
**B.6.** **To organize our replies, we split this comment into three sections:**
   a) **"P.6, L.11-14, this sentence caught my interest, assuming that microwave remote sensing is sensitive to the very first cm of soil, at least at X- and C-band, could you detect all irrigation types? Is dripping/micro-irrigation leading to a sufficient change in soil moisture to be noticed from space?**
   b) **What about vegetation masking the ground? We all have in mind pictures of irrigated (fully developed) corn, can remote sensing see through that? Although it is mentioned in 5.1.3 maybe it could ruled out C-band? At least for certain period of vegetation cycle.**

**c) By the way, why not considering SMOS L-band mission, my understanding is that it should be more sensitive to soil moisture than the sensors you use (?). SMAP is used but in combination with SMOS (I am aware that combined products exist) it would lead to a longer period being investigated."**

**Reply:**

**a.)** This is a crucial question. In the manuscript, we tried to stress the fact that different irrigation techniques are expected to critically impact the sensitivity of microwave observations. For instance, see Section 2.1, Page 7, Lines 31-32 expanding to Page 8, Line 1: "It is likely that the sensitivity of satellite soil moisture retrievals to irrigation increases when the irrigation application efficiency of a particular irrigation system or technique deteriorates. Therefore, we expect higher sensitivity towards gravity- (e.g. flood and furrow irrigation), and lower sensitivities towards sprinkler- and micro-irrigation systems." This dependency should also vary between L- and C-band, however we argue that the ability to sense "the very first centimeter of soil" is too pessimistic. With respect to irrigation, we are not aware of any observationally driven studies investigating this issue in-depth, so we don't know for sure what is the actual case. However, in [1], drip, sprinkler, and flood irrigation parameterizations are assessed in a modelling domain. Regarding your question, we think that the most important points made in this study are:

   i) Even in developed countries such as the U.S., micro-irrigation is very rare (e.g., 1% of farmers in the Central Plains use it to an unknown extent), since it is more cost and labour intensive. Therefore, it is very likely that it only accounts for a very small portion of the overall water usage.

   ii) The top-panel in Figure 2 shows "monthly, domain-averaged differences from Control in top-layer soil moisture SM" for each irrigation method. The authors state that: "As anticipated, Drip exhibits zero changes to soil moisture content because of the nature of the algorithm, as additional water is immediately used for transpiration." In real-world examples, this budget-closure likely won't be the case, but certainly very close to it. Here, one has to differentiate between surface- and subsurface-drip-irrigation. Sub-surface irrigation at the plants root zone has the goal of not even wetting the soil, so the impact on soil moisture is negligible. For surface drip-irrigation, the soil wetting effect will probably be very minor.

   iii) They also state that although soil moisture is not impacted directly by drip irrigation, the method does have an effect on soil temperature and hence on latent heat flux and surface temperature. So although soil moisture is not directly affected by drip irrigation in the model output, there is considerable impact on the energy- and water-cycle. However, this is not the main subject of our study.

Based on these observations, we argue that the impact of micro-irrigation (such as drip irrigation) on space-born soil moisture observations will likely be below the retrieval error, if measurable at all.

**b.)** This is another important point. The reality is that we are not completely sure what is the case. Specifically at the start of the study, we had problems with the active ASCAT soil moisture product over the Corn Belt (see [2]), which according to USGS accounts for the majority of U.S. corn production. We figure that during the latter stages of crop development, when corn reaches significant heights (which in the end can potentially be in the order of 2.5-3m), the microwave signal will still partly reflect soil moisture. As outlined in the reply to comment A.1., ASCAT soil moisture was repeatedly shown to perform well over densely vegetated areas. However, we also expect that volume scattering effects (in the active sensing case) and vegetation water content (in the passive sensing case) might increasingly affect the overall signal. In the end, this is more related to the vegetation correction of each respective retrieval algorithm and hence is out of the scope of this study.

**c.)** Concerning the use of SMOS soil moisture, we also kindly refer to our replies to comment A.1. Concerning the use of a combined SMAP-SMOS soil moisture product, we agree that this could be a very valuable addition, but advocate that before using multi-sensor data (e.g., ESA CCI soil moisture, SMAP-SMOS soil moisture data records), we should assess the specific sensitivities of each sensor with respect to irrigation quantification.

**Changes in manuscript:**
None

* * *
**B.8.** "P.6, L.32, "[...] that SMAP soil moisture carries a clear irrigation signal from rice irrigation [...]", could you please specify irrigation type, flooding (as it is likely to be water seeded)? Did SMAP sees open-water there?"

**Reply:**
As this statement directly refers to the study conducted by Lawston et. al., 2017 (who used the SMAP product sampled to a 9 km grid), we cannot make a definite statement. However, since part of our time series analysis is based on the same study region (Sacramento Valley, California), we are confident that SMAP sensed open water during a certain period of time.

**Changes in manuscript:**
None
* * *
**B.9.** "P.11, L.4, didn't you say earlier that this map was not reliable? Please clarify."

**Reply:**
We agree that in the current state, the discussion of the ESA CCI land cover product is misleading. To clarify, this product proves to provide a very reliable indicator for agricultural land cover in general, but shows a low performance in further classifying irrigated and non-irrigated agriculture. In addition, it is subject to misclassification in other parts of the world (see [1]). Since we are only interested in mask for agricultural land in general, we can use this map. As you point out, in Section 1.2.1, Page 5, Lines 2-4, we make the case that "...the irrigated class is likely to be considered unreliable…".

**Changes in manuscript:**
In order to eradicate potential confusion, we propose to clarify our choice in Section 3.3, Page 12, Line 4: "Despite the drawbacks discussed in section 1.2, the ESA CCI Land Cover data set (Bontemps et al., 2013) was used to create a cropland mask for the CONUS, because the classification of overall agricultural land cover (i.e., irrigated and rainfed lands) proved to be accurate."

In addition, we modified Section 1.2.1, Page 5, Lines 2-4. It now reads as follows:
"It distinguishes irrigated and non-irrigated cropland for 2000, 2005 and 2010. However, we argue that over the contiguous United States (CONUS) the irrigated class is likely to be considered unreliable, as apparently all irrigated lands are wrongly attributed to the non-irrigated agriculture class."
* * *
**B.10.** **"Section 3.4, then considering SMOS would make it possible investigating longer period of time."**
**Reply:**
Since the first reviewer also commented on SMOS, we already addressed the topic in the replies to the first reviewer. Therefore, we kindly refer to our reply on comment A.1.
* * *
**B.11.** **"P.11, L.18-23, please see work from Kumar et al., 2015 on the scaling issue."**

**Reply:**
The scaling issue was addressed in comment A.2. of the other reviewer. We kindly refer to our reply above.
* * *
**B.12.** **To organize our replies, we split this comment into five sections:**
   a) **"P.12, sections 4.1 to 4.3, so a big assumption is that the mismatch between model and satellite soil moisture is irrigation, so the forcing is assumed to be perfect and we know that is not the case.**
   b) **I am also curious on the possible mismatch between what land cover the satellite is really sensing and what MERRA2 has for land cover.**
   c) **The same is true for soil texture, porosity and all ancillary data...could it lead to spurious irrigations/non-irrigation?**
   d) **And what about temporal mismatch? Do you consider the satellite soil moisture revisit time enough for such study? If it rains after the satellite has passed (but maybe it a silly thought as you consider rain free period -according to the forcing)? Please comment on this issue.**
   e) **You should also assess your method in areas where crops are rain-fed only to see what signal is detected when we know that no irrigation occurs (see 5.1.3!)"**

**Reply:**
   a) Yes, this is the main assumption. By applying the mean-stdv scaling, we implicitly also correct for deviations in spatial representativeness etc., as discussed in the reply to comment A.03. Of course, the forcing is not perfect. Since rainfall is arguably the most important model forcing over the land surface, we use information from an additional CPC precipitation data set at 0.25° resolution. This basically allows us to have a 4x better spatial resolution when evaluating whether or not an irrigation event is present compared to the native rainfall forcing in MERRA-2. However, we do not gain more resolution with respect to soil moisture and ultimately irrigation estimation.
   b) We agree that this is an important point. In addition, while the model land cover usually remains static/semi-static, the satellite microwave observations are constantly sensitive to changes in land cover. However, such an evaluation would require a reliable land cover reference data set and would imply an actual land cover validation of the MERRA2 land surface model, which is beyond the scope of this study. Maybe this issue can be addressed in future studies.

c) Mismatches in ancillary data such as soil parameters would imply a bias in the soil moisture dynamic range of the model. Since we scale the satellite data to this dynamic range, we implicitly account for such mismatches.

d) The satellite revisit time arguably has a big impact. In [1], our co-authors conducted a synthetic experiment to analyze the impact of revisit time (among other factors such as retrieval error and climate) on irrigation quantification. They found that the performance of their method significantly deteriorates for revisit times > 3 days. Concerning our study, as outlined in Section 4.2, Page 13, Lines 10-14, this is an intrinsic limitation of the method: "If an irrigation event is detected during an observation gap of > 4 days, we check if there has been a significant increase in the model soil moisture (e.g. due to rainfall) within that period. When more than one significantly positive model slopes (or precipitation events) occur during the gap-period, we cannot say for sure if the observed increase in soil moisture was due to irrigation or precipitation and therefore conservatively disregard the potential irrigation event."

e) In each of the four focus regions, parts of the "non-irrigated" reference pixels are rainfed crops (see section 5.3), so we think that this is already being taken into account. Of course, we had to make a trade-off between geographical proximity (very similar climate, crops, etc…) and a large fraction of rainfed cropland.

**Changes in manuscript:**
We tried to address comments a) - c) by extending and re-writing the discussion on spurious irrigation events in section 4.3, page 15:

"Potential errors may arise when the model forcing misses or creates false rainfall events. In addition, because of differences in timing of the estimates and differences in represented soil depth between remotely sensed and modelled soil moisture, their response to precipitation events may differ as well. This can lead to spurious irrigation events when irrigation is estimated at days with rainfall. Therefore, we use information from an additional CPC precipitation data set at 0.25° resolution, thus providing approximately 4x higher spatial resolution than the rainfall product used to force MERRA-2 (see section 3.2). This allows us to make a more educated guess when evaluating if the observations and/or model estimates are affected by rainfall.

Furthermore, if a potential irrigation signal coincides with preceding rainfall we assume that irrigation is unlikely and disregard the event. In some extreme cases, capillary rise from deeper soil layers or run-on can wet the top soil. Theoretically, these conditions are reflected by the satellite soil moisture retrievals, but absent in the model soil moisture simulations (i.e. if such effects are not accounted for in the soil hydrology formulation of the LSM (McColl et al., 2017)). However, at the large spatial scales represented by the employed satellite (approximately 25 km) and model soil moisture products (approximately 50 km), very 5 few pixels are expected to show positive $\Delta\Theta sat$ or $\Delta\Theta mod$ in the absence of precipitation or irrigation.

Another impact concerns mismatches between the ancillary data used to force the model and parametrize the respective satellite soil moisture retrieval algorithms, such as land cover and soil parameters. By rescaling to the model dynamic soil moisture range, we implicitly account for mismatches in soil parameters. However, addressing potential mismatches in land cover was out of the scope of this paper and thus represent an intrinsic limitation of the method."

**Reply:**
Please see our reply to your comment B.6.(a).
* * *
**B.15.** **"Pleas reshape figure 6 as text is hardly readable (and label panels as much as possible for sack of clarity"**

**Reply:**
We thank the reviewer for the suggestions for improving the readability of figure 6 and have reorganized it.

**Changes in manuscript:**
Page 37, Figure 6: Page 37, Figure 6
We have increased the font sizes and deleted the county labels for improved readability.

[revised manuscript text omitted]